# A Fine Regulation of the Hippocampal Thyroid Signalling Protects Hypothyroid Mice against Glial Cell Activation

**DOI:** 10.3390/ijms231911938

**Published:** 2022-10-08

**Authors:** Lamis Chamas, Isabelle Seugnet, Roseline Poirier, Marie-Stéphanie Clerget-Froidevaux, Valérie Enderlin

**Affiliations:** 1CNRS/MNHN UMR 7221-Phyma “Physiologie Moléculaire et Adaptation” Department of “Life Adaptations” Muséum National d’Histoire Naturelle 57, rue Cuvier CP 32, CEDEX 05, 75231 Paris, France; 2Institut des Neurosciences Paris-Saclay, Université Paris-Saclay, CNRS, 91400 Saclay, France

**Keywords:** hypothyroidism, hippocampus, microglia, astrocytes

## Abstract

Adult-onset hypothyroidism is associated with learning and cognitive dysfunctions, which may be related to alterations in synaptic plasticity. Local reduced levels of thyroid hormones (THs) may impair glia morphology and activity, and promote the increase of pro-inflammatory cytokine levels mainly in the hippocampus. Given that neuroinflammation induces memory impairments, hypothyroidism-related glia dysfunction may participate in brain disorders. Thus, we investigated the mechanisms linking hypothyroidism and neuroinflammation, from a protective perspective. We induced hypothyroidism in adult C57BL/6J and wild-derived WSB/EiJ male mice by a seven-week propylthiouracil (PTU) treatment. We previously showed that WSB/EiJ mice were resistant to high-fat diet (HFD)-induced obesity, showing no neuroinflammatory response through adaptive abilities, unlike C57BL/6J. As PTU and HFD treatments are known to induce comparable inflammatory responses, we hypothesized that WSB/EiJ mice might also be protected against hypothyroidism-induced neuroinflammation. We showed that hypothyroid WSB/EiJ mice depicted no hippocampal neuroinflammatory response and were able to maintain their hippocampal thyroid signalling despite low circulatisng TH levels. In contrast, C57BL/6J mice exhibited disturbed hippocampal TH signalling, accompanied by neuroinflammation and memory impairment. Our results reinforce the preponderance of the hippocampal TH regulatory system over TH circulating levels in the hippocampal glial reactivity.

## 1. Introduction

Thyroid hormones (THs), particularly thyroxin (T4) and triiodothyronine (3,3′,5-triiodothyronine, T3), regulate major physiological functions and are particularly powerful modulators of brain functions throughout life (reviewed in [1,2]). TH actions are predominantly mediated by T3 binding to nuclear thyroid hormone receptors (TRs), which act as transcriptional factors regulating the expression of specific thyroid hormone-responsive genes [3]. T4 is the major TH secreted by the follicular cells of the thyroid gland, whereas T3, the transcriptionally active TH, is mainly produced locally by deiodination of T4 by the type-2 iodothyronine deiodinase (DIO2). Circulating T4 and to a lower extent, T3 reach the brain by crossing the blood-brain barrier via specific transporters. T4, which has a higher affinity for transporters, enters the astrocytes via transporters, where it is deiodinated by DIO2 to produce T3 (reviewed in [4,5]). Subsequently, T3 is released from the astrocytes and captured by neurons via distinct transporters. Neurons are not able to directly generate T3 because they do not express DIO2. Though, they are mainly dependent upon astrocytes for their T3 supply, even if some T3 can also directly come from the circulation [3,5].

DIO2 is highly expressed in the brain throughout life. Moreover, the type-3 deiodinase (DIO3) also modulates T3 brain availability by converting both T4 and T3 into inactive metabolites ([6], reviewed in [7]). Thus, whereas DIO2 contributes to T3 availability under euthyroid conditions and helps to maintain the brain T3 concentration, particularly in hypothyroidism conditions, DIO3 provides an additional regulatory mechanism to control brain T4 and T3 levels. Indeed, growing evidence suggests that serum TH may not always accurately reflect the hormonal status of target tissues ([8,9], reviewed in [10]). Regarding the brain, a slight variation from the normal/homeostatic range of brain T3 may have widespread effects on the central nervous system [1]. This emphasises the importance of a local sensitive regulatory system, implying not only the interaction between deiodinases but also the other actors of thyroid signalling as the TRs.

In humans, hypothyroidism during the foetal and postnatal periods, if untreated, leads to intellectual deficiency as well as neurological, cognitive and motor deficits [11]. Hypothyroidism in adulthood also has significant mental consequences such as anxiety, depressive symptoms and cognitive deficits (reviewed in [12]), which may be related to changes in synaptic plasticity [13]. Additionally, in adult, THs affect glia morphology and therefore may affect brain function, leading to influence on neuron-glia interaction (reviewed in [14]). THs also modulate microglial activity at the cellular level by being responsible for microglia activation, cell migration and phagocytosis [15]. Hypothyroidism is also associated with astrogliosis, significantly higher levels of C-reactive protein markers, and proinflammatory cytokines [16,17,18]. The hippocampus, a key structure involved in spatial learning and memory, is highly sensitive to TH levels due to a high expression of TRs [3]. Untreated hypothyroid patients and rodents showed a decrease in hippocampal volume [19]. Moreover, studies on hypothyroidism have reported an impairment of hippocampal mechanisms important for synaptic plasticity and memory [20,21]. Moreover, an increase in pro-inflammatory cytokines has been observed in the hippocampus of hypothyroid rodents, in addition to cognitive dysfunctions [18,22]. Interestingly, TH supplementation of hypothyroid animals was able to restore memory function and decrease neuroinflammatory markers [23,24]. Even if the effects of hypothyroidism on neuroinflammatory mechanisms remain very poorly understood, data from the literature report that adult hypothyroidism leads to a wide range of molecular dysfunctions in the hippocampus converging on plasticity and memory impairments.

Given that neuroinflammation induces cognitive and memory impairments [25,26,27], hypothyroidism-related glial dysfunction may participate in brain and mental disorders (reviewed in [4]). In the present study, our objective was to achieve new insights into the link between hypothyroidism and neuroinflammation. To this end, we compared the response to induced hypothyroidism in two strains of mice: one laboratory strain, the C57BL/6J strain, and one wild-derived strain, the WSB/EiJ strain. The wild-derived mice are good models to understand human phenotypic diversity and are commonly used in comparison studies with other mouse strains for this purpose [28]. Regarding the WSB/EiJ mice, they are genetically distinct from other mouse strains, including C57BL/6J, and possess genomic variations (new alleles) that could be the source of the observed phenotypic divergences, such as lower body weight and resistance to diet-induced obesity. Moreover, despite lower T4 circulating levels than most other mouse strains, including the C57BL/6J strain, WSB/EIJ mice seem to be adapted to live with (https://phenome.jax.org/strains/42; accessed on 16 August 2022).

Thus, we induced hypothyroidism in adult C57BL/6J and WSB/EiJ male mice by administering the antithyroid molecule propylthiouracil (PTU) for seven weeks through feeding. We have previously shown that the euthyroid WSB/EiJ mice depict a strong high-fat diet (HFD)-induced obesity resistance: they don’t gain weight after 3 days or 8 weeks of HFD, and consequently, don’t show any neuroinflammatory response to HFD in the hypothalamus region [29], which is described for a long time as associated to obesity [30]. In contrast, C57BL/6J mice are prone to obesity in response to HFD and develop hypothalamic neuroinflammation. Our previous results suggested that the resistance of WSB/EiJ mice could be, at least in part, explained by differences in mitochondrial reactivity and differential expression of genes involved in inflammatory and mitochondria pathways in response to HFD, unlike C57BL/6J mice [29]. THs are largely implicated in metabolic homeostasis and major interactions have been reported between thyroid function, weight control and obesity (reviewed in [31,32]). In addition, Muthu and collaborators revealed in 2018 a comparable effect of PTU treatment and HFD on the development of inflammatory responses and a synergic effect with both treatments [33]. Thus, we hypothesized that WSB/EiJ mice could be protected against hypothyroidism-induced neuroinflammation, as observed under the HFD diet. Deciphering the mechanisms responsible for this could pave the way to understanding protective processes against the establishment of neuroinflammation.

In this study, we investigated some indicators of local T3 availability (T3 target genes and genes involved in TH signalling) and markers of astrocytes and microglia activation in the hippocampus of C57BL/6J and WSB/EiJ male mice treated with PTU. Hippocampal-dependent learning and memory were also explored in mice. We showed that the hypothyroid WSB/EiJ mice depicted no hippocampal neuroinflammatory response and were able to maintain their hippocampal thyroid signalling despite low circulating TH levels. In contrast, hypothyroid C57BL/6J mice exhibited disturbed hippocampal TH signalling, accompanied by neuroinflammation. In addition, they displayed spatial memory impairment.

## 2. Results

### 2.1. Hypothyroid Status

We first evaluated thyroid status between the two mouse strains after 7 weeks of PTU treatment by measuring circulating T4 levels (Figure 1). Euthyroid WSB/EiJ mice depicted a lower circulating total T4 [3.4 μg/dL] than C57BL/6J mice [6.1 μg/dL] (*p* = 0.0003), but still in the euthyroid reference range for mice (see discussion section for details). In response to PTU treatment, circulating levels of total T4 decreased by 90% and 82% in C57BL/6J and WSB/EiJ mice [0.6 μg/dL], respectively, compared to their euthyroid controls (*p* < 0.001). These results confirmed that both strains presented peripheral hypothyroidism of an equivalent level after 7-weeks of PTU treatment.

We then evaluated thyroid status in the hippocampus of the two mouse strains after seven weeks of PTU treatment by measuring the mRNA expression of TH signalling genes (TH receptors (*Trα1* and *Trβ1* isoforms; Figure 2a), deiodinases type 2 and 3 (*Dio2*, *Dio3;*
Figure 2b)) and TH target genes (*Klf9*, *Rc3*; Figure 2c), usually used as biomarkers of local T3 availability [34,35,36]. We observed that PTU treatment induced a differential effect in the mRNA expression of these genes between the two mouse strains (strain x treatment interaction *p* < 0.01, see Appendix A). Gene expression of these biomarkers was overall downregulated in the hypothyroid C57BL/6J mice compared to their controls (0.0001 < *p* < 0.05). However, their expression was unchanged or upregulated in WSB/EiJ mice in response to PTU treatment. Together, these data suggest that circulating hypothyroidism could induce a decrease in T3 cellular actions in the hippocampus of C57BL/6J mice whereas, in WSB/EiJ mice, adaptive mechanisms could take place to maintain T3 availability in this brain region.

### 2.2. Consequences of Hypothyroidism on Hippocampal Inflammation

Neuroinflammation was evaluated in the mouse hippocampus. We first measured by western-blot the expression of STAT3 and its phosphorylated form (p-STAT3), considered as a central regulator of glial reactivity (Figure 3a). Relative STAT3 expression was globally higher in C57BL/6J hippocampus than in WSB/EiJ (Strain effect *p* = 0.007) and remained unchanged after PTU treatment in the hippocampus of both mouse strains (treatment effect *p* > 0.05). Whereas, the expression of p-STAT3 only increased in C57BL/6J hippocampus in response to hypothyroidism (strain x treatment interaction *p* = 0.006). p-STAT3/STAT3 ratio was increased only in the hippocampus of the C57BL/6J mice in response to PTU treatment (strain x treatment interaction *p* = 0.003) by 171% compared to their euthyroid group (*p* = 0.026), whereas no significant change was observed in the hypothyroid WSB/EiJ mice (*p* > 0.05). We next assessed phenotypic modifications of astrocytes (Figure 3b,c) and microglia (Figure 4). We quantified the density of the astrocytes population (GFAP+ cells) within the hippocampal CA1 region, a brain area involved in synaptic plasticity and spatial memory mechanisms [37] (Figure 3b,c). We first observed that astrocyte density was considerably lower (by 40%) in WSB/EiJ CA1 than in C57BL/6J strain (Strain effect *p* = 0.0001). Furthermore, the density of GFAP+ astrocytes was increased in the hypothyroid C57BL/6J mice (by 27% compared to euthyroid mice, *p* = 0.0001). In contrast, the density of GFAP+ astrocytes was maintained in the hypothyroid WSB/EiJ mice compared to their euthyroid group (*p* > 0.05), and at a much lower level than the hypothyroid C57BL/6J mice (*p* < 0.001).

Finally, we examined the expression of hippocampal IBA1 and CD11B proteins as markers of microglia activation. Our results showed that IBA1 expression remained statistically unchanged in the hippocampus of both strains in response to PTU treatment (*p* > 0.05) (Figure 4a). Nevertheless, in the hypothyroid C57BL/6J mice, we observed an increase in IBA1 level (about 28%) compared with the controls. CD11B expression (Figure 4b) was also significantly increased (strain x treatment interaction *p* = 0.006) in the hypothyroid C57BL/6J mice (+96%) compared to their euthyroid group (*p* = 0.0005); No change was observed in WSB/EiJ mice (*p* > 0.05). In the CA1 hippocampal region (Figure 4c,d), a strain difference was first reported, with an overall lower microglial density in WSB/EiJ strain than in C57BL/6J strain (Strain effect *p* = 0.0001). The density of IBA1+ microglia, in response to PTU treatment, was different between the two strains (strain x treatment interaction *p* = 0.041): it was slightly increased by about 17% in C57BL/6J mice but without reaching the significance (*p* > 0.05) but sparsely reduced in WSB/EiJ mice by 9% compared to their euthyroid group (*p* = 0.04). To test the effect of PTU treatment on microglial cell activation, we performed double immunostaining for CD68/IBA1 in the CA1 hippocampal region (Figure 4e,f). 3D Imaris quantification revealed a significant increase in the volume ratio of CD68/IBA1 labelling in PTU-treated C57BL/6J mice (strain x treatment interaction *p* < 0.00001) by approximately 60% compared to their euthyroid controls (*p* = 0.0001) (Figure 4f), whereas this ratio was unchanged in the hypothyroid WSB/EiJ compared to the euthyroid group (*p* > 0.05) (Figure 4f). Taken together, these results showed that hypothyroidism leads to microglia activation in the hippocampus of C57BL/6J mice but not in WSB/EiJ mice.

### 2.3. Hypothyroidism Effect on Behavior and Cognition

#### 2.3.1. Open-Field Test

Spontaneous locomotor activity was analysed during the exploration of an open-field (OF) arena for 15 min (Figure 5). The two mouse strains behaved differently in this test, as observed by higher total distance travelled and velocity in WSB/EiJ mice compared to C57BL/6J (Strain effect: 0.0001 < *p* < 0.05; Figure 5a). Moreover, both strains responded differently to PTU treatment (strain x treatment interaction: 0.01 < *p* < 0.05): while PTU-treated C57BL/6J velocity was not altered (*p* > 0.05), it was significantly reduced in the hypothyroid WSB/EiJ mice (*p* = 0.006).

To evaluate anxiety-like behavioural responses, we analysed the travelled distance and the time spent in the center zone of OF arena: results also revealed differences between strains (strain x treatment interaction: 0.01 < *p* < 0.05; Figure 5b). In euthyroid condition, the WS/EiJ mice travelled less distance, spent less time and had higher velocity in the centere than euthyroid C57BL/6J mice (*p* < 0.05), suggesting an anxiety-like behaviour of WSB/EiJ mice. Interestingly, PTU treatment did not modify these three parameters in C57BL/6J mice (*p* > 0.05), indicating no hypothyroid-related effect on anxiety-related responses in this strain. In contrast, the PTU-treated WSB/EiJ mice spent more time, travelled more distance and had a lower velocity in the centre compared to their euthyroid group (*p* < 0.05; Figure 5b). Together, these results reflected an anxiety-like behaviour of WSB/EiJ mice that is reduced by PTU treatment.

#### 2.3.2. Barnes Maze Test

To evaluate the impact of hypothyroidism on spatial learning and memory, a Barnes Maze (BM) test was conducted for both strains. For the euthyroid C57BL/6J mice, the travelled distance and latency to enter the escape hole progressively decreased over the training days, indicating that mice learnt the location of the escape hole (*p* < 0.05; Figure 6a,b). Although the travelled distance of the hypothyroid C57BL/6J mice decreased for the first two days (*p* < 0.05), similar to the euthyroid group, they showed a stagnation in the distance between D2 and D4 (*p* < 0.05), and this difference was no longer observed at the end of the acquisition phase (D5). Regarding the velocity, no difference was observed between both the euthyroid and hypothyroid C57BL/6J mice over the acquisition days (*p* > 0.05; Figure 6c), suggesting that the differences in the distance and latency observed between the two groups were not due to altered locomotor activity.

During the probe test (Figure 6d), the euthyroid C57BL/6J mice spent more time in the target quadrant (45.88 [30.32–61.85] s) than in the other quadrants (around 23 s in each adjacent quadrant and 6.68 [0–14.96] s in the opposite) (*p* < 0.05). The time spent in the target quadrant was significantly higher than the 25% chance level (*p* = 0.006), meaning that they learned to localize and remember the location of the escape box. In contrast, the hypothyroid C57BL/6J mice spent the same time in each quadrant (around 24 s), which was not different from the 25% due to chance (*p* > 0.05). Compared to the euthyroid mice, the PTU-treated mice spent significantly less time in the target quadrant and more in the opposite quadrant, indicating spatial memory impairment in response to hypothyroidism (*p* < 0.01). These results were in line with the number of errors (Figure 6e) made by the euthyroid C57BL/6J mice that were statistically more important in the target than in the other quadrants (*p* < 0.05), indicating their persistency in searching for the escape box in the correct quadrant as they memorized the escape hole location. On the other hand, the hypothyroid mice made the same number of errors in all quadrants (*p* > 0.05) (fewer errors in the target and more in the opposite quadrants compared to the euthyroid mice, *p* > 0.05), confirming that they did not memorize the escape hole location and performed a random search in the maze.

Taken together, the BM test results reveal learning delay in the PTU-treated C57BL/6J mice associated with spatial memory deficit in response to hypothyroidism. Regarding the WSB/EiJ strain, the results obtained during the acquisition phase of functional assays showed that spatial memory cannot be assessed using the Barnes test (Appendix A). Indeed, the mice moved very quickly towards the maze’s periphery, without using distal cues to locate the escape hole. In addition, they showed no interest in the escape hole since they could get in and out very quickly.

## 3. Discussion

Several studies report an association between hypothyroidism and cognitive disorders, but the mechanisms are currently poorly understood. Based on our previous data [29], we hypothesized that the WSB/EiJ mice could be protected from hippocampal low TH availability and consequently from neuroinflammation as observed under the HFD diet, as they are naturally adapted to live with lower circulating T4 levels than most of the mouse strains (Donahue 1 study, https://phenome.jax.org/measures/11531; accessed on 16 August 2022). The present comparative study shows a close correlation between disturbed hippocampal TH signalling and neuroinflammation. Moreover, glial activation was associated with spatial memory dysfunction in the C57BL/6J mice.

Despite the lower T4 circulating levels of WSB/EIJ compared to the C57BL/6J mice, this strain is still considered euthyroid as its T4 levels are included in the euthyroid reference range for mice [2.6–9 μg/dL] established by the comparison of 28 non-pathological mouse strains with no sign of hypothyroidism (Donahue 1 study, https://phenome.jax.org/measures/11531; accessed on 16 August 2022). If we do the analogy with humans, a T4 level is considered normal, which means, not inducing any symptoms of hypothyroidism, when it is between 5.0 to 12.0 μg/dL. It is therefore not abnormal to have a variation of two-fold between two different individuals, and that does not mean that the individual with the lower level is hypothyroid, as long as it is inside the normal range. Moreover, as described in our former study, young euthyroid WSB/EiJ mice presented an increase in their body weight similarly to C57BL/6J mice from one month to 4 months of age, reflecting a normal growth period and well-being status of young euthyroid adult mice [29]. We induced hypothyroidism in mice with a low iodine diet supplemented with PTU, a worldwide-recognized protocol to induce hypothyroidism in rodents. It has been well established that PTU treatment reduces both T4 and T3 concomitantly. For example, Cortes and collaborators reported that PTU treatment for 20 days induced a significant decrease level of serum T3 and T4 levels [17] as also shown by Sener et al., 2006 [38]: in the PTU-treated rats the serum T3 and T4 levels (20.8 ± 0.22 and 0.31 ± 0.01 ng/dL) were found to be significantly lower than those of the control group (55.3 ± 0.63 and 6.810.32 ng/dL, *p* < 0.001). Other authors also attest to circulating hypothyroidism testing only T4, as Artis et al., reported that serum T4 levels in comparable PTU treated rats were lower than the control values (0.248 ± 0.038 μg/dL vs 1.276 ± 0.578 μg/dL, [21]. Vallortigara et al., confirmed hypothyroidism by a drastic reduction in T4 levels (by - 96%) in PTU/MMI-treated mice compared with controls [39]. A comparable decrease in free-T4 levels in serum was observed in thyroidectomized rats [23]. Therefore, based on these numerous data, in the present study, we only assayed T4 and not T3 to verify that PTU treatment had induced circulating hypothyroidism in our treated mice of both strains. Indeed, we confirmed a drastic reduction in T4 levels in both PTU-treated mouse strains as compared to controls, thus validating the circulating hypothyroidism.

Knowing that T3 is the main transcriptionally active TH, we next assessed hippocampal T3 availability by measuring the main actors in TH signalling as well as the regulation of T3 target genes, as already widely described in the literature [34,35,36,40,41]. In the hippocampus, we observed a decrease in T3 target gene expression only in the C57BL/6J mouse strain. The local action of iodothyronine deiodinases contributes to about 50% of T3 produced in postnatal and adult rodents [42]. In the hypothyroid C57BL/6J mouse strain, the *Dio2* mRNA level was unchanged in the hippocampus, and the *Dio3* mRNA level was reduced. DIO2, as an activating deiodinase, favours the presence of T3 by converting T4 into T3, whereas DIO3 reduces T3 levels, converting T3 into T2. Thus, our results suggest a decrease in T3 local availability in the hippocampus, as *Dio2* expression, as not increased, would not allow the compensation of the lower T4 levels entering the cells, due to the circulating hypothyroidism. The decrease in *Dio3* expression also confirms that T3 levels are very low in these cells. This hypothesis was confirmed by the lower expression of T3 target genes, attesting that PTU treatment reduced the brain cellular action of T3 in the hippocampus of the C57BL/6J mouse strain. These last results are in line with our previous data obtained in adult PTU-treated rats [22] and in PTU/MMI-treated mice [39].

In contrast, in the WSB/EiJ strain, we observed a strong increase in the amount of hippocampal *Dio2* mRNA level, allowing an increased activity of this enzyme to de-iodinate T4 in astrocytes and maintain the T3 availability in the hippocampus. T3 availability was also attested by the target gene expression regulation, which was at least maintained (*Rc3*)*,* if not increased as *Klf9* or *Trβ1* in the PTU-treated WSB/EiJ mice. This compensatory mechanism could protect the mouse brain against the deleterious effects of peripheral hypothyroidism. Our data reinforce the crucial role of DIO2 in the brain as a protective enzyme rescuing from peripheral TH deficiency [5], and once again demonstrate that the circulating TH levels do not necessarily reflect the brain T3 availability [8] and that the local TH signalling is physiologically and functionally more relevant than circulating TH levels.

Glial cells, particularly astrocytes and microglia, are considered key regulatory elements in the functional plasticity of neural networks. The progression of several neurological diseases depends on glial reaction that promotes inflammatory processes [43]. Neuroinflammation contributes to cognitive dysfunctions [26]. Our previous data reported an increased level of proinflammatory cytokines associated with PTU treatment [22]. Here, we aimed to examine the effect of PTU treatment on glial cell activity and assess the strength of evidence for a link between hypothyroidism and glia activation. Thus, some molecular markers of astrocyte and microglia activation were evaluated to determine glial reactivity in our experimental model. Multiple pathways are associated with glial reactivity. Among them, the JAK/STAT3 pathway appears as a central player in the induction of glial reactivity (reviewed in [44,45,46]). In healthy conditions, STAT3, a member of the Janus-kinase-STAT (JAK/STAT3) signalling family, remains in an inactive cytoplasmic conformation. Then, upon activation, STAT3 translocates into the nucleus where it behaves as a transcription factor. This activation of STAT3 is induced by phosphorylation on a critical tyrosine residue (Tyr705) and the dimerization of the protein: its activation by phosphorylation is described as increased after CNS insults [47]. Our Western-blot analysis revealed an increase of STAT3 phosphorylation only in the hippocampus of the hypothyroid C57BL/6J strain. Comparable results have already been obtained by Millot and collaborators in the hippocampus of LPS-treated mice compared to their controls [48].

One of the best-known target genes of STAT3 is GFAP, commonly used as a marker of astrogliosis and astrocyte activation in several situations involving brain injury [49]. A feature of reactive astrocytes across different species is the upregulation of GFAP level and the increase in the number of astrocytes stained by GFAP [17,48,49,50,51]. In the present work, the PTU-treated C57BL/6J mice also displayed an increase in GFAP+-cell density in the CA1 hippocampal region. No modification was observed in the PTU-treated WSB/EiJ mice. Regarding microglia, when they are activated, they move to sites of injury where they can recruit or activate other cells, but also proliferate, phagocytize debris, and promote reorganization of neurons in altered brain areas (reviewed in [52]). Microglia activation is accompanied by morphological changes. In healthy environments, the majority of microglia have ramified process morphology with multiple branches that extend as sensors for potential challenges: this is the resting state. Chronic inflammatory state leads first to a hypertrophic branched morphology with an enlarged soma, and then cells become amoeboid-like, with enlarged soma and fewer branches and processes; these two states are accompanied by the release of pro-inflammatory cytokines [53]. Among microglia markers, IBA1 is commonly used to mark all microglia: its expression is upregulated during activation [54]. Microglia can also express a typical immune-inflammatory profile characterized by higher levels of CD11B and CD68 proteins [52]. Indeed, CD11B is one of the first indices of microglia activation as they prepare to adhere to damaged cells. CD68 is a glycoprotein on the lysosomal membrane of macrophages and microglia, which is indicative of phagocytic activity related to amoeboid morphology [55]. As for astrocytes, our results clearly showed a microglial response related to PTU treatment, but only in the hippocampus of the C57BL/6J strain. IBA1 protein expression and cell density were increased by 28% and 17%, respectively, in the hypothyroid C57BL/6J mice compared to their control group. Interestingly, our data also revealed a significantly increased expression of CD11B protein levels in the hippocampus, associated with an increase of CD68/IBA1 volume in the CA1 of these mice. Thus, our results indicated that PTU treatment favour glia activation in the hippocampus of the C57BL/6J mice. Therefore, we can speculate that the strong reactive profile of microglia observed only in the C57BL/6J PTU-treated mice would be associated with disturbed hippocampal TH signalling. These data are in agreement with those in the literature reporting a critical role of T3 on morphogenic effects on microglia (reviewed in [4]). Furthermore, in vitro, exposure to T3 can promote the survival of purified microglia cells and elongation of their processes [56].

Thus, the discrepancy in glial reactivity between the C57BL/6J and the WSB/EiJ strains consolidates the link between T3 brain content and neuroinflammation: the local compensatory mechanism of peripheral hypothyroidism via the increase of *Dio2* expression would protect the brain of WSB/EiJ mice against low T3 availability and thus, against central inflammation.

Various studies of cognitive functions in animal models of hypothyroidism have reported spatial memory deficits [23,57,58]. In this study, we used the Barnes maze test, a hippocampal-dependent spatial memory task [59,60] to examine the memory performance of PTU-treated mice. Our findings revealed an alteration in spatial memory in the hypothyroid C57BL/6J mice showing a transient delay in the learning phase and deficit in spatial long-term memory. These results are in line with our previous findings reporting that PTU treatment results in deficits in hippocampal-dependent spatial memory [22]. As described above, hypothyroidism in C57BL/6J mice leads to activation of microglia and astrocytes known to interact with neurons and consequently control brain function. The literature revealed that complex neuron-glia networks promote plasticity-related processes in neurons where both types of neural cells work in concert, serving different activities (reviewed in [14,50]). In neuroinflammatory conditions, the delicate balance between the various neuro-glia interactive components that regulate normal brain functioning is interrupted leading to impaired neural plasticity and consequently memory (reviewed in [25]). In this way, Liu and collaborators showed a negative correlation between *Cd11b* or *Gfap* mRNA levels and synaptic proteins in the hippocampus, which suggests that the excessive activation of microglia and astrocytes may contribute to decreased synaptic proteins in the hippocampus [61]. Thus, the strong glial reactivity observed in the PTU-treated C57BL/6J mice could be one of the key mechanisms promoting memory impairment. Literature data reported a clear correlation between RC3 level, a brain-specific gene involved in synaptic plasticity and memory via the modulation of Ca2+/calmodulin-dependent signalling [62], and cognitive function. For example, ageing and hypothyroidism are conditions that are associated with spatial memory decline and with a decreased level of *Rc3* [62,63]. In Morris’s test, RC3^-/-^ mice are unable to establish the search criterion to reach the submerged platform and they swim without showing clear signs of orientation (reviewed in [64]). Thus, the decreased amount of hippocampal *Rc3* observed in PTU-treated C57BL/6J mice let us suggest that glial reactivity induced by hippocampal TH hypo-signalling might alter neuron-glia interaction disrupting synaptic plasticity and thus memory performance. Our results are consistent with those revealing, in postnatal rats, that microglia is an important partner in hypothyroidism-induced hippocampal neuronal loss and learning-memory impairment [65].

Regarding the WSB/EiJ strain, our data revealed that the Barnes test did not allow us to assess their spatial memory. These results are consistent with recent data showing that WSB/EiJ mice were not able to perform the spatial recognition task using the Y-maze test: euthyroid WSB/EiJ male mice did not demonstrate a preference for the new arm compared to the familiar arm [66]. These authors explained that the WSB/EiJ strain was significantly more active and stressed than the euthyroid C57BL/6J mice. This interpretation is in line with the higher velocity and the anxiety-like behaviour we observed in the WSB/EiJ compared to the C57BL/6J control mice in the OF test. Interestingly, other authors demonstrated that mice with astrocyte-specific *Dio2* inactivation exhibit anxiety-depression-like behaviour [67]. Finally, a reduced expression of *Dio2* and *Dio3*, as we observed in the euthyroid WSB/EiJ mice compared to the euthyroid C57BL/6J mice, has also been reported in stressed mice [68]. This is further demonstrated by the increase of hippocampal *Dio2* expression and the decrease of the anxiety-like behaviour of the hypothyroid WSB/EIJ mice. In this study, we also carried out preliminary experiments in euthyroid WSB/EiJ male mice using another test, the spatial object recognition test. Unfortunately, WSB/EiJ mice were too active to explore objects, and during the retention session, they were not spending more time exploring the moving object. Thus, this test was discontinued revealing again the anxiety-like phenotype of the WSB/EiJ strain. All these data from the literature and our experiments allow us to suggest that the inability of WSB/EiJ mice to perform the Barnes test could be due to their anxiety-like phenotype. As described previously, we showed that the fine regulation of the T3-hippocampal signalling in the PTU-treated WSB/EiJ mice protects them from hippocampal low TH availability and consequently, from neuroinflammation. Our results also reported a similar *Rc3* mRNA level in the hypothyroid WSB/EiJ mice and the euthyroid C57BL/6J mice. Thus, although it is necessary to develop WSB/EiJ strain-specific cognitive assays, all these data allow us to hypothesize that circulating hypothyroidism would not affect neuron-glia interaction thus, not hippocampal functions.

## 4. Materials and Methods

### 4.1. Animals, Treatment, and Sample Collections

Animals were housed individually under a 12:12 light-dark cycle (07h00–19h00), maintained at 23 °C, with food and drinking water provided ad libitium. Wild-type C57BL/6J and WSB/EiJ breeder mice were purchased from Charles River (Saint-Germain-sur-l’Arbresle, Rhône France) and Jackson Laboratories (Bar Harbor, Maine, USA), respectively. Hypothyroidism was induced in 8-weeks old male mice by given an iodine-deficient diet supplemented with 0.15% propylthiouracil (PTU) (1.5 g PTU/kg diet, Envigo Teklan, Madison, WI, USA) for 7 weeks as described in [22,24]. This antithyroid molecule blocks the activity of thyroid peroxidases that catalyse the iodination of thyroglobulin and are essential for thyroid hormone synthesis. Moreover, PTU also inhibits DIO1, which produces T3 by deiodination of T4 in peripheral tissues, such as liver, or kidney, and therefore reduces the release of T3 in the blood [69], but has no action on local DIO2 activity [70]. Euthyroid control (CTRL) mice were fed with a standard chow diet. After 7 weeks of treatment, retro-orbital blood (n = 7–8 per group) was collected with heparinized haematocrit capillary tubes (Schott Hirschmann, Iéna, Germany) in the morning just before mice’s euthanasia by decapitation. Blood samples were allowed to clot for 1 h at room temperature (RT), centrifuged (15 min, 3000 × g, RT) and serum supernatants stored at −20 °C. Hippocampus samples (n = 6–9 per group) were dissected and snap frozen in liquid nitrogen and stored at −80 °C until further analysis (Western-blot and RNA analysis). To collect brain samples for the immunohistochemistry experiment (n = 3 per group), mice were given a no-recovery dose of anaesthetic and then were perfused with PBS followed by paraformaldehyde 4%, and then the fixed brain was collected and processed as described in 4.4. Body weight and food intake were monitored twice a week during the treatment, as markers of well-being. A total of max 12 animals/group were used in this study. All procedures were conducted according to the principles and procedures in Guidelines for Care and Use of Laboratory Animals and was validated (68.096) by the MNHN ethical comity for animal experimentations.

### 4.2. Circulating Thyroid Hormone Levels

Serum total T4 concentrations (n = 7–8 per group) were measured using an ELISA kit (Labor Diagnostika Nord (LDN), Nordhorn, Germany) according to the manufacturer’s instructions.

### 4.3. Reverse-Transcription qPCR

Total RNA from hippocampus samples (n = 6 per group) was extracted using RNABle lysis reagent (Eurobio, Les Ulis, France) and the RNeasy^®^ Mini Kit (Qiagen, Les Ulis, France) according to the manufacturer’s protocol. All RNA samples were quantified using Qubit 2.0 Fluorometer (Invitrogen life technologies, Villebon-sur-Yvette, France)) and RNA integrity was evaluated using Agilent 2100 Bioanalyzer. Complementary DNA (cDNA) synthesis was performed using Reverse Transcription Master Mix from Fluidigm^®^ according to the manufacturer’s protocol with random primers. Real-time quantitative PCR was carried out with QuantStudio 6 Flex Real-Time PCR System (Applied Biosystems) using TaqMan Universal PCR master mix (Applied Biosystems, Villebon-sur-Yvette, France) and pre-designed TaqMan probes (TaqMan Gene Expression Assays, Applied Biosystems). Gene expression assays: Iodothyronine Deiodinase 2 (*Dio2*, Mm00515664_m1); Iodothyronine Deiodinase 3 (*Dio3*, Mm00548953_s1); Thyroid hormone receptor alpha (*Trα1*, Nm178060.3); Thyroid hormone receptor beta (*Trβ1*, Mm01316711_m1); Kruppel-like factor 9 (*Klf9*, Mm00495172_m1); Neurogranin (*Rc3*, Mm01178296_g1); Hypoxanthine guanine phosphoribosyl transferase (*Hprt*, Mm00446968_m1); Actin beta (*Actb*, Mm00607939_s1). The RT-qPCR reaction for each sample was conducted in duplicates and direct detection of the PCR product was monitored by measuring the increase in fluorescence generated by the TaqMan probe. The qRT-PCR data were analysed using QuantStudio™ Real-Time PCR Software (version 1.3, Thermo Fisher Scientific) and ExpressionSuite software (version 1.0-4, Life technologies, Villebon-sur-Yvette, France). Two housekeeping genes (*Actb*; *Hprt*) were selected based on Vandesompele et al. method [71] and using the SlqPCR package (version 1.42.0). A custom R tool was constructed to measure relative gene expression levels according to the ΔΔCT method [72] and to perform non-parametric statistical tests (two-way ANOVA with permutation) as described previously [29]. Graphical representations (boxplot whiskers) were performed using FC values (fold-changes compared to C57BL/6J CTRL) on GraphPad Prism (GraphPad Software Inc., San Diego, CA, USA; version 8.0-2).

### 4.4. Immunohistochemistry

Hippocampal immunohistochemistry (n = 3 per group) was assayed as described previously [29] with minor changes. Briefly, brains were fixed with paraformaldehyde 4%, cryoprotected in 30% sucrose, embedded in Frozen Sections Compound (Leica Biosystems) and stored at −80°C. Thirty μm thick coronal brain sections were cut using a cryostat (Leica, Nanterre, France). Five floating sections per mouse were incubated in a blocking solution composed of 10% normal goat serum (NGS; Sigma, Saint-Quentin Fallavier, France) and 1% bovine serum albumin (BSA; Sigma) diluted in PBS for 1 h at room temperature (RT). Hippocampal sections were incubated with the following primary antibodies: rabbit anti-IBA1 (1/750, Wako Chemicals, Sobioda, Montbonnot-Saint-Martin, France), chicken anti-GFAP (1/300, Abcam, Paris, France), or rat anti-CD68 (1/1000, Abcam) and secondary antibodies: donkey anti-rabbit (Alexa Fluor 488, 1/500, Invitrogen, Villebon-sur-Yvette, France), donkey anti-chicken (Alexa 594 nm, 1/500, Invitrogen) or donkey anti-rat (Alexa 594 nm, 1/500, Invitrogen). Brain sections were observed under TCS-SP5 Leica confocal microscope using x400 magnification. Acquisitions were obtained using max intensity Z projection of 30 μm-thick z-stacks (a minimum of 20 images with a z-step of 1.01 μm) for each hippocampal CA1 region of interest (ROI; left and right side). Quantifications of microglia and astrocytes were performed with ImageJ software (National Institutes of Health, USA) using the cell counter plugin. Results of GFAP and IBA1 densities are presented as the number of immuno-positive cells per mm2 for each ROI. Three-dimensional confocal images for the double staining IBA1/CD68 were reconstructed using Imaris software (Bitplane, Zurich, Switzerland). To measure and visualize the colocalization of CD68 and IBA1, a volume filter was first applied to remove nonspecific staining. Then, a “shortest distance” filter was applied to remove CD68 volume that was not within the microglial volume. Activated microglia were determined by calculating the ratio of CD68/IBA1 volumes.

### 4.5. Western Blot Analysis

Frozen mouse hippocampal samples (n = 6–9 per group) were lysed mechanically using Tissue Lyzer (Qiagen, Courtaboeuf, France), and proteins were extracted in RIPA buffer (Sigma-Aldrich, Saint-Quentin Fallavier, France) supplemented with protease and phosphatase inhibitors (Pierce Protease and Phosphatase Inhibitor Mini Tablets, Thermo Scientific™). Protein concentrations were determined by bicinchoninic acid method using the Pierce™ BCA Protein Assay kit (Thermo-Fisher Scientific, Les Ulis, France). Total protein lysates (30 μg) were resolved by SDS-PAGE using stain-free precast gels (4–15% Mini-Protean Tgx Stain-Free Protein Gels, Biorad, Marnes-la-Coquette, France). Gels were then activated by Ultraviolet light with the ChemiDocTM Touch Imaging System (Biorad) prior to protein transfer onto a nitrocellulose membrane (Biorad). Membranes were blocked with 5% BSA or free-fat milk in Tris-/Phosphate-buffer saline (TBS/PBS)-Tween 20 (0.1%, Sigma) followed by overnight incubation at 4°C with the primary antibodies: anti-CD11B (1:1000, Abcam, Paris, France), anti-IBA1 (1:850, Wako-Sobioda, Montbonnot-Saint-Martin, France), anti-Phopho-STAT3 (Tyr705) (1:1000, Cell Signalling Technology), anti-STAT3 (1:3500, Cell Signalling Technology). After three washes with the appropriate PBS/TBS-T, membranes were incubated with the anti-rabbit IgG Peroxidase conjugate secondary antibody (1:3000, Sigma-Aldrich) and immunocomplexes were visualized by enhanced chemiluminescence (Clarity™ Western ECL Substrate, Biorad) according to manufacturer’s instructions and using ChemiDoc system. We used stain-free total protein measurement as the loading control [73]. Densitometry of bands was quantified with Image Lab software (Biorad, Marnes-la-Coquette, France). Target protein density values were first normalized to total protein content of their respective lanes and the ratio was then normalized to the density of the internal reference sample that was run in each gel in parallel of samples to avoid gel to gel variability. For all groups, relative expression of each target protein was compared to the C57BL/6J CTRL group.

### 4.6. Behavioural Test

Behavioural testing was undertaken after 6 weeks of treatment between 9 am and 7 pm after daily handling for 1 week.

#### 4.6.1. Open Field

The open field test (OF) was performed to evaluate the locomotor activity and the anxiety-like behaviour of mice (n = 8 per group). Mice were placed onto the centre of the OF (40 cm width × 40 cm length × 50 cm height) and observed for 15 min. The behaviour of each mouse was monitored with an infrared camera (T840, FLIR systems) and automatically analysed using a video tracking system (Ethovision XT 14, Noldus Information Technology, Wageningen, The Netherlands). For analysis, the apparatus was divided into two zones, i.e., the center (the most anxiogenic area; 20 cm × 20 cm) and the peripheral zone (least anxiogenic area). Total travelled distance, velocity and distance and time spent in the center zone were recorded.

#### 4.6.2. Barnes Maze

Spatial learning and memory performances were tested in a Barnes Maze (n = 6/7 per group for C57BL/6J mice and 3/4 for WSB/EiJ). Regarding the C57BL/6J strain, we started the experiment with 8 mice per group. Two euthyroid mice were unable to enter the escape box and were therefore removed from the experiment. Similarly, one hypothyroid mouse was eliminated by Dixon’s test because it differentially performed than its group in the probe test. The maze consisted of a circular white PVC platform (100 cm diameter) elevated 100 cm from the floor and containing twenty equally spaced holes (diameter = 5 cm) located on the periphery. A black escape box (22.9 × 5.3 × 8.6 cm) was mounted underneath the platform covering one single hole designated as the escape hole. The remaining holes were blocked with an identical cylinder without an escape box. Extra-maze visual clues of different geometric shapes were positioned on the walls of the room and were kept fixed throughout the test. An aversive bright light (1000 lux) was used to motivate the mouse to search for the escape box. The protocol was composed of two phases: the acquisition and the probe phases. In the acquisition phase, mice were subjected to two trials per day, one in the morning and one in the afternoon, for 5 consecutive days. During this phase, the mouse was placed in the centre of the maze and was allowed to explore the maze for 300 s searching for the escape hole. If the latency to enter the escape hole exceeded 300 s, the mouse was guided toward it. After entering, the mouse was left in the escape box for 1 min. On the probe day, 48 h after the last acquisition day, the escape box was removed and blocked identically to the other holes. During this phase, the mouse was placed in the centre of the maze and was allowed to explore the maze for 60 s. A video camera placed above the centre of the maze was used to monitor performances. For the acquisition phase, distance travelled, velocity and latency to escape into the box were analysed using Ethovision software. For the probe phase, time spent in each quadrant was measured as well as the number of errors for the 60 s. An error was counted when the mouse’s nose poked the non-target hole while searching for the escape box. For these analyses, the maze was virtually divided into 4 quadrants: the target quadrant in which the escape box was located, adjacent right, adjacent left (clockwise from the target quadrant) and opposite. Each quadrant was considered as 25% of the total maze.

### 4.7. Statistical Analyses

Unless specifically mentioned, all data were performed using non-parametric statistical tests (two-way ANOVA test with permutations) as described previously [29]. Dixon’s Q-test was used for identification and rejection of outliers. Differences were considered significant for a *p*-value ≤ 0.05. All data are represented as medians using boxplot and min-max whiskers on GraphPad Prism (GraphPad Software Inc., San Diego, CA, USA; version 8.0-2). Concerning the Barnes Maze test, to evaluate learning performances during the acquisition days, two-way repeated measures ANOVA followed by Tukey’s multiple-comparisons test were carried out using GraphPad prism. Dixon Q-test was applied for the time spent in each quadrant data and statistical analysis of one sample t-test was applied using StatView software (SAS Campus Drive, Cary, North Carolina, USA; Version 5.0). The chance level was fixed at 25% so if the percentage of time spent in quadrants was higher than 25%, then this was considered not hazardous. For the time spent in each quadrant, data are represented in stacked bars and expressed as (median [±Confidence interval 95%]).

## 5. Conclusions

In conclusion, the main aim of the present work was to achieve new insights into the relationship between hypothyroidism and neuroinflammation. Altogether, our data showed that the fine regulation of the hippocampal TH signalling in the PTU-treated mice protects them from glial reactivity and neuroinflammation. Moreover, our data obtained from C57BL/6J mice reinforces the link between neuroinflammation and memory deficits. Unfortunately, the problem encountered handling WSB/EiJ mice in the memory test prevented us from determining the link between TH local availability and spatial memory capacity (Figure 7). Interestingly, the present study also demonstrated that WSB/EiJ mice constitute a particularly valuable experimental model since these mice have a notable adaptive capacity allowing them to cope with both metabolic and thyroid dysregulations, via compensatory mechanisms. Indeed, WSB/EiJ mice can maintain metabolic homeostasis as well as hippocampal T3-signalling during metabolic and thyroid disorders respectively, supporting a close relationship between thyroid status and metabolic functions. Given the occurrence of hypothyroidism in obese or diabetes patients, and that low T3 brain levels are associated with various cognitive declines, further research is needed to understand the relationship between metabolic homeostasis processes and the brain regulation of TH content to allow for the development of preventive measures to improve cognitive health.

## Figures and Tables

**Figure 1 ijms-23-11938-f001:**
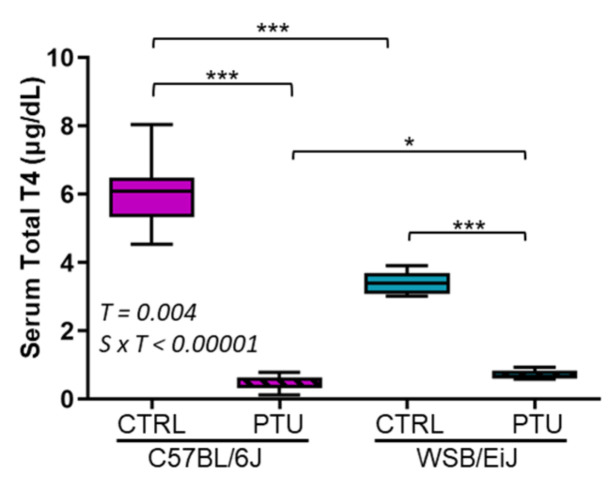
**PTU induces hypothyroidism in both mouse strains.** Serum total T4 levels were significantly reduced after 7 weeks of PTU treatment in C57BL/6J and WSB/EiJ mice compared to the euthyroid control mice (CTRL) (n = 7–8 per group, non-parametric two-way ANOVA with permutation tests). Statistically significant effects with the respective *p*-values are indicated on the graph (T: treatment, S × T: strain x treatment interaction). Box plots represent median values and min-max whiskers. Post Hoc test results are shown in the graph (*, *p* ≤ 0.05; ***, *p* ≤ 0.001).

**Figure 2 ijms-23-11938-f002:**
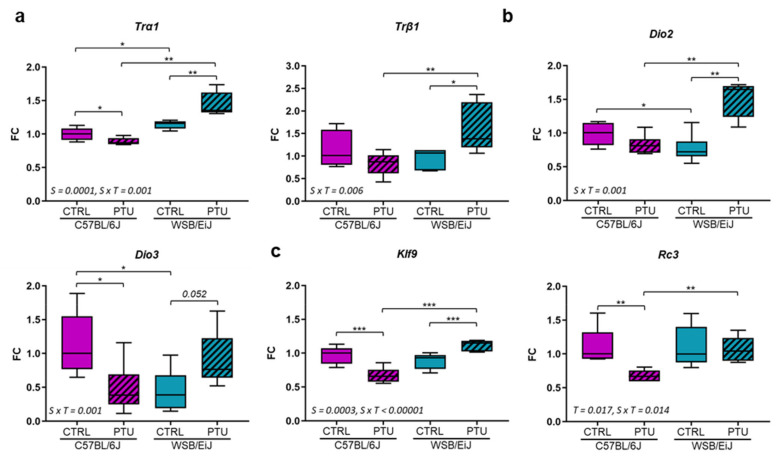
**Thyroid signalling in the hippocampus of the two strains of mice after PTU treatment:** Expression of TH receptors *Trα1* and *Trβ1* (**a**), deiodinases *Dio2* and *Dio3* (**b**) and TH target genes *Klf9* and *Rc3* (**c**). TH signalling and TH target genes were overall downregulated in PTU-treated C57BL/6J mice but upregulated or unchanged in PTU-treated WSB/EiJ mice. Data are represented as relative fold-change in expression (FC). Box plots represent median values and min-max whiskers. Statistically significant effects (non-parametric two-way ANOVA with permutation tests) with the respective *p*-values are indicated on the graph (S: strain effect, T: treatment effect, S × T: strain × treatment interaction). Post Hoc tests results are indicated on the graph (n = 5–6 per group; *, *p* ≤ 0.05; **, *p* ≤ 0.01; ***, *p* ≤ 0.001).

**Figure 3 ijms-23-11938-f003:**
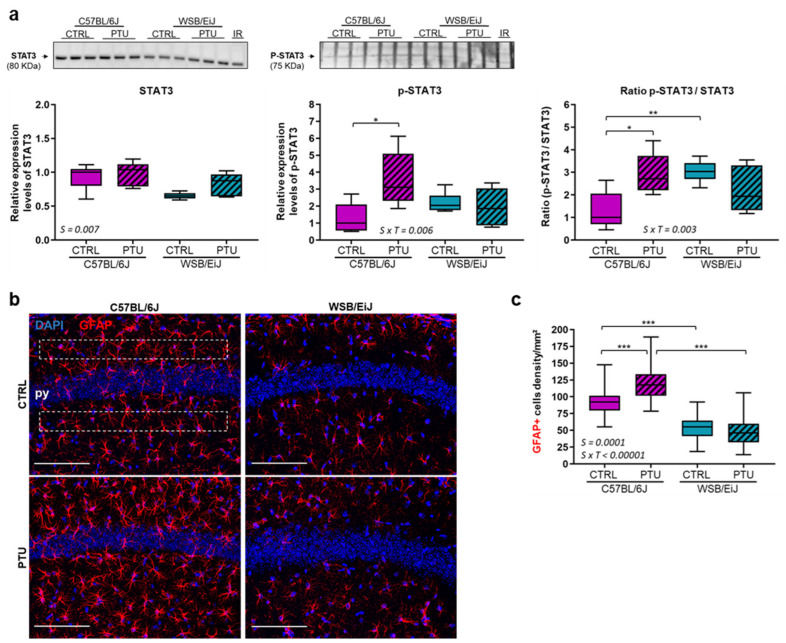
**Neuroinflammatory status in the hippocampus of the two strains of mice after PTU treatment** (**a**) (Upper panel) Representative western blot images of total STAT3 and its phosphorylated form p-STAT3 expressed in the hippocampus of euthyroid (CTRL) and hypothyroid (PTU) C57BL/6J and WSB/EiJ mice (n = 3 per group/western-blot). Densitometry values for each protein band were normalized to the internal reference sample (IR). (Lower panel) Densitometry analysis of the relative expression (to C57BL/6J CTRL group) indicated that total STAT3 expression remained unchanged in both strains whereas p-STAT3 expression was increased in the hippocampus of C57BL/6J mice after PTU treatment. Ratio (p-STAT3/total STAT3) showed an increase of STAT3 phosphorylated form in the hippocampus of the hypothyroid C57BL/6J mice whereas it stayed unchanged in the hypothyroid WSB/EiJ mice (n = 5–6 per group) (**b**,**c**) Astrocyte density is increased in the hippocampal CA1 of the hypothyroid C57BL/6J mice, but not in the hypothyroid WSB/EiJ mice: (**b**) Representative confocal images of GFAP+ astrocytes (red) in the CA1 hippocampal region of euthyroid (CTRL) and hypothyroid (PTU) C57BL/6J (left panel) and WSB/EiJ (right panel) mice. Cell nuclei are stained with DAPI (blue). py: pyramidal cell layer. Scale bars = 100 μm. (**c**) Both strains showed a significant differential response to PTU treatment: the density of GFAP+ astrocytes is increased in the CA1 of C57BL/6J mice, whereas unchanged in the CA1 of WSB/EiJ mice in response to PTU-treatment. There was a lower density of astrocytes in the CA1 of WSB/EiJ mice compared to C57BL/6J mice, occurring independently of the PTU treatment (n = 3 mice per group, n = 4 sections per mouse and n = 4 ROI (white dashed square) per section). Boxplots represent median values and min-max whiskers. Statistically significant effects (non-parametric two-way ANOVA with permutation tests) with the respective *p*-values are indicated on the graph (S: strain effect, S × T: strain × treatment interaction). Post Hoc test results are indicated in the graph (*, *p* ≤ 0.05; **, *p* ≤ 0.01; ***, *p* ≤ 0.001).

**Figure 4 ijms-23-11938-f004:**
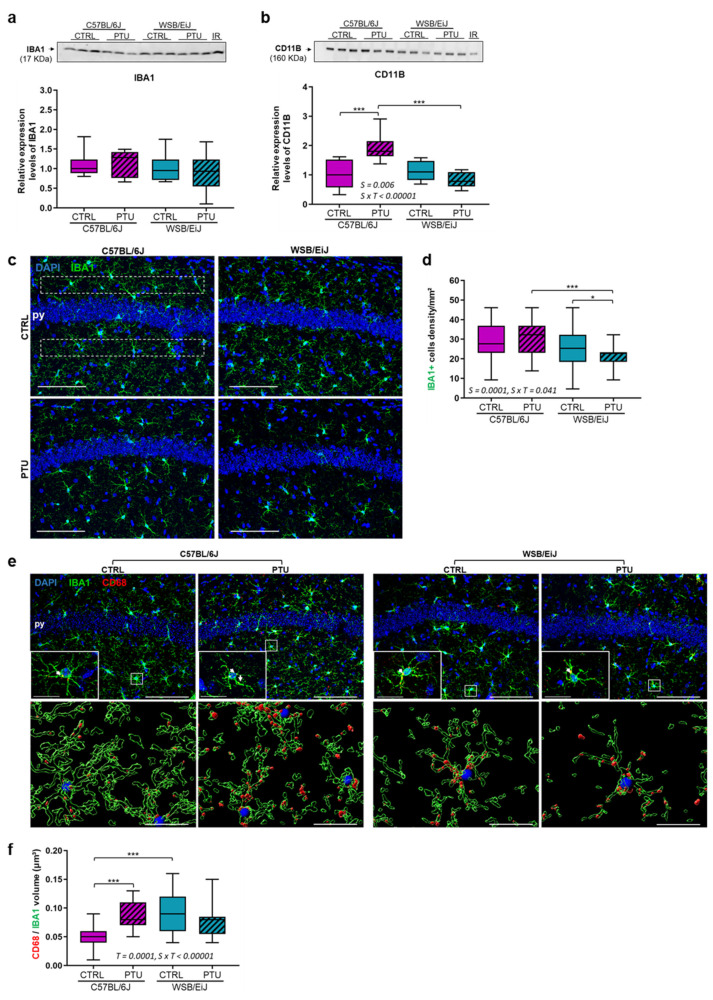
**Microglia cells are activated in the hippocampal CA1 of C57BL/6J mice in response to PTU, but not in WSB/EiJ mice**. ((**a**,**b**), upper panel) Representative western blot images of IBA1 (**a**) and CD11B (**b**), an activated microglia marker, expressed in the hippocampus of euthyroid (CTRL) and hypothyroid (PTU) C57BL/6J and WSB/EiJ mice (n = 3 per group/western-blot). Densitometry values for each protein band were normalized to the internal reference sample (IR). (**a**,**b**, lower panel) Relative expression (to C57BL/6J CTRL group) of IBA1 was unchanged in the hippocampus of both strains, whereas CD11B expression was increased only in C57BL/6J hippocampus after PTU treatment (n = 8–9 per group). (**c**) Representative confocal images of IBA1+ microglia (green) in the CA1 hippocampal region of euthyroid (CTRL) and hypothyroid (PTU) C57BL/6J (left panel) and WSB/EiJ (right panel) mice. Cell nuclei are stained with DAPI (in blue). White dashed square: ROI; py: pyramidal cell layer. (**d**) The density of IBA1+ microglia was unchanged in C57BL/6J mice whereas decreased in the hippocampal CA1 of WSB/EiJ mice in response to PTU-treatment; a strain effect revealed a lower density of microglia in the hippocampal CA1 of WSB/EiJ mice than in C57BL/6J mice. (**e**) (Upper panel) Representative confocal images of IBA1+ (in green) and CD68+ (in red) activated microglia within CA1 hippocampal region of euthyroid (CTRL) and hypothyroid (PTU) C57BL/6J and WSB/EiJ mice. Cell nuclei are stained with DAPI (in blue). Inserts showed details of IBA1+ CD68+ activated microglia (white rectangle). The colocalization of CD68 in the microglia cell (yellow staining) is identified by the white arrowhead. Scale bars: 100 μm in main images and 20 μm in inserts. (Lower panel) Three dimensional (3D) images of magnified CA1 hippocampal region show the colocalization of IBA1+ microglia (green) with CD68 (red). Scale bars: 20 μm. (**f**) Volume quantification analysis revealed an increase of (CD68 volume/IBA1 volume) ratio in the CA1 of C57BL/6J mice, whereas it was unchanged in WSB/EiJ mice in response to PTU-treatment (n = 3 mice per group, n = 4 sections per mouse and n = 4 ROI per section). For all graphs of the figure, boxplots represent median values and min-max whiskers. Statistically significant effects (non-parametric two-way ANOVA with permutation tests) with the respective *p*-values are indicated on the graphs (S: strain effect, T: treatment effect, S × T: strain × treatment interaction). Post Hoc tests results indicated on the graphs: *, *p* ≤ 0.05; ***, *p* ≤ 0.0001.

**Figure 5 ijms-23-11938-f005:**
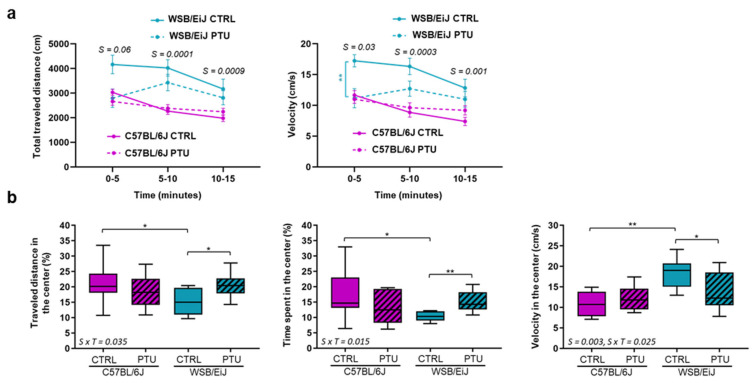
**Exploration and anxiety-related behaviours in the OF are affected only in the WSB/EiJ mice following PTU-treatment**. (**a**) Total travelled distance and velocity were measured in the OF arena every 5 min for 15 min, after six weeks of PTU treatment for euthyroid (CTRL), hypothyroid (PTU) C57BL/6J, and WSB/EiJ mice (n = 8 per group). The data are expressed as mean ± SEM. Statistically significant strain effects with the respective *p*-values are indicated on the graph (non-parametric two-way ANOVA with permutation tests was applied for each time point; S: strain effect). Post Hoc tests results are indicated on the graph (**, *p* ≤ 0.001). (**b**) The travelled distance, the velocity and the time spent in the center of the open-field for 15 min were measured after six weeks of PTU-treatment for euthyroid (CTRL), hypothyroid (PTU) C57BL/6J, and WSB/EiJ mice (n = 8 per group). No effect of PTU was observed in C57BL/6J mice, whereas the hypothyroid WSB/EiJ mice spent more time and travelled more distance in the center, with a reduced velocity, than the euthyroid group. Boxplots represent median values and min-max whiskers. Statistically significant effects with the respective *p*-values (non-parametric two-way ANOVA with permutation are indicated on the graph (S: strain effect, S × T: strain × treatment interaction). Post Hoc tests results are indicated on the graph (*, *p* ≤ 0.05; **, *p* ≤ 0.01)).

**Figure 6 ijms-23-11938-f006:**
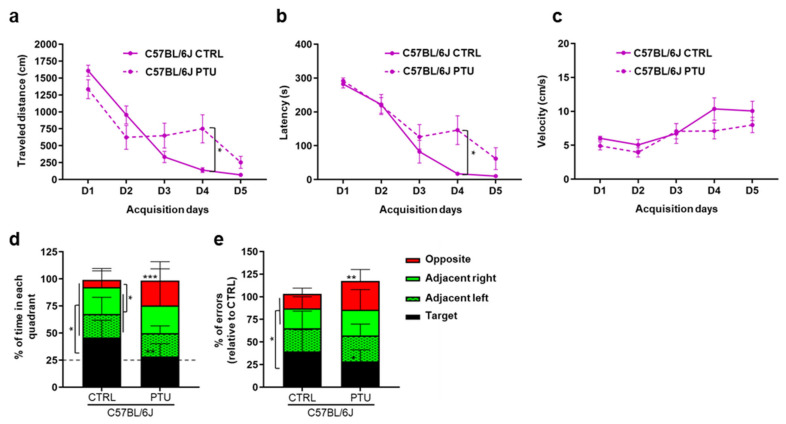
**Spatial learning and memory are altered in C57BL/6J mice after PTU-treatment.** Measurement of the travelled distance (**a**) latency (**b**) and velocity (**c**) to escape into the box of the Barnes maze during the five acquisition days, evaluating learning performances of euthyroid (CTRL) and hypothyroid (PTU) C57BL/6J mice (n = 6–7 per group). Statistical differences between treatment and time were assessed by two-way repeated measures ANOVA followed by Tukey’s multiple-comparisons test. The data are expressed as mean ± SEM. Spatial memory was assessed by measuring the percentage of time spent (**d**) and the number of errors made (**e**) in each quadrant of the maze for 60 s on the Barnes maze probe test day (48 h after the last acquisition day). The horizontal line (in grey) indicates the time in a zone expected by chance (25%). The hypothyroid C57BL/6J mice spent less time and did fewer errors in the target quadrant than their euthyroid controls. Post Hoc test results are indicated on the graph (non-parametric two-way ANOVA with permutation tests; *, *p* ≤ 0.05; **, *p* ≤ 0.01; ***, *p* ≤ 0.001). Stacked bars represented as median [±Confidence interval 95%].

**Figure 7 ijms-23-11938-f007:**
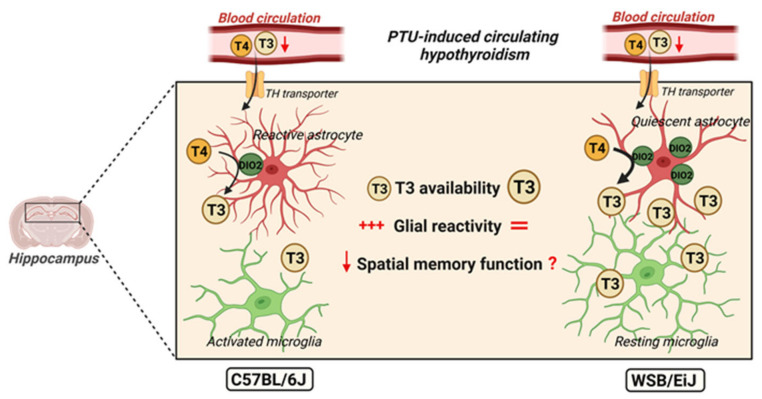
**Potential mechanisms linking hippocampal TH hyposignalling and neuroinflammation, as observed comparing C57BL/6J and WSB/EiJ mouse strains**. Circulating hypothyroidism was induced by PTU treatment, leading to a lesser amount of T3 and T4 reaching the hippocampus by crossing the BBB via specific transporters. Hypothyroid C57BL/6J mice exhibited disturbed hippocampal TH signalling (revealing reduced T3 availability) accompanied by neuroinflammation (astrocyte reactivity and microglia activation). In contrast, hypothyroid WSB/EiJ mice depicted no hippocampal neuroinflammatory response and were able to maintain their hippocampal thyroid signalling as attested by target gene expression despite low circulating TH levels, notably via an increase in *Dio2* expression. The spatial memory test was completed only in C57BL/6J mice: data obtained in this strain reinforce the link between neuroinflammation and memory deficits. *This figure was created with BioRender.com.*

## Data Availability

Not applicable.

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
