# Peer review of "A Fine Regulation of the Hippocampal Thyroid Signalling Protects Hypothyroid Mice against Glial Cell Activation"

_ijms, 2022, doi:10.3390/ijms231911938_

Round 1
Reviewer 1 Report (Previous Reviewer 1)
Journal: IJMS (ISSN 1422-0067)
Manuscript ID: ijms-1895532
Type: Article
Title: A fine regulation of the hippocampal thyroid signalling protects hypothyroid mice against glial cell activation.
Section: Molecular Endocrinology and Metabolism
Special Issue: Molecular Mechanism of Hypothyroidism
This manuscript aims to study relationship between hypothyroidism in mice and hippocampal glial cell activation, neuroinflammation as well as learning and memory deficits. The presentation of the manuscript is not good. I have the following comments and suggestions for the authors to improve the quality of the manuscript.
1. What are the differences between WSB/EiJ mice and C57BL/6J? Add some introduction in the revised manuscript.
2. L 113-114
“Euthyroid WSB/EiJ mice depicted a lower circulating total T4 [3.4 µg/dL] than C57BL/6J mice [6.1 µg/dL] (p= 0.0003).”
WSB/EiJ mice depicted a lower circulating total T4 [3.4 µg/dL] than C57BL/6J mice [6.1 µg/dL] (p= 0.0003). Are WSB/EiJ mice euthyroid? How do you define euthyroid?
3. Add analysis of serum T3 concentrations.
4. L 329-335
“Knowing that T3 is the main transcriptionally active TH, we assessed the the hippocampal T3 availability by measuring main actors in TH signalling as well as the regulation of T3 target genes, as already widely described in the literature [33-35,38]. In the hippocampus, we observed a decrease in T3 target gene expression only in the C57BL/6J mouse strain. The local action of iodothyronine deiodinases contributes to about 50% of T3 produced in postnatal and adult rodents [40].”
Insert the reference 39.
5. L 353-408
This paragraph is too long. Divide it to two paragraphs or more.
6. Section “4.1. Animals,treatment, and sample collections”
How many mice were used in this study? Add the information in the revised manuscript.
Author Response
1.What are the differences between WSB/EiJ mice and C57BL/6J? Add some introduction in the revised manuscript.
The main difference between WSB/EiJ and C57BL/6J mice are that the first one is a wild-derived inbred mouse strain (https://phenome.jax.org/strains/42), whereas C57BL/6J mice are laboratory inbred mice. The wild-derived mice are good models to understand the human phenotypic diversity, and they
are commonly used in comparison studies with other mouse strains to understand phenoptypic differences Keanes et al., 2011 (ref 28 in the manuscript). Regarding the WSB/EiJ strain, WSB/EiJ are genetically distinct from other mouse strains, including C57BL/6J, and possess genomic variations (new
alleles) that could be the source of the observed phenotypic divergences, as lower body weight andresistance to diet-induced obesity (Lee et al, 2011, doi: 10.1210/en.2011-0060).
We now introduce this strain in the manuscript with the following sentence (line 82):
“To this end, we compared the response to induced-hypothyroidism in two strains of mice: one laboratory strain, the C57BL/6J strain, and one wild-derived strain, the WSB/EiJ strain. The wildderived mice are good models to understand the human phenotypic diversity, and are commonly used in comparison studies with other mouse strains in this purpose [28]. Regarding the WSB/EiJ mice,
despite lower T4 circulating levels than most other mouse strains, including the C57BL/6J strain, they seem to be adapted to live with (https://phenome.jax.org/strains/42).”
2. L 113-114
“Euthyroid WSB/EiJ mice depicted a lower circulating total T4 [3.4 µg/dL] than C57BL/6J mice [6.1 µg/dL] (p= 0.0003).”WSB/EiJ mice depicted a lower circulating total T4 [3.4 µg/dL] than C57BL/6J mice [6.1 µg/dL]
(p= 0.0003). Are WSB/EiJ mice euthyroid? How do you define euthyroid?
As reported in the manuscript (lines 322 to 325), the T4 levels of WSB/EiJ mice is included in the euthyroid reference range for mice [2.6-9 µg/dL] established by the comparison of 28 nonpathological mouse strains [Donahue 1 study https://phenome.jax.org/measures/11531]. If we do the analogy with Human, a T4 level is considered normal in Human when it is between 5.0 to 12.0μg/dL. It is therefore not abnormal to have a variation of two-fold between two different individuals, and that does not mean that the individual with the lower level is hypothyroid, as long as it is inside the normal range. This sentence was added to the manuscript to illustrate the euthyroid state of the WSB/EiJ mice.
Moreover, WSB/EiJ mice do not present any signs of hypothyroidism. Actually, PTU treatment is known to arrest growth of animals as hypothalamo-pituitary-thyroid (HPT) axis is a key actor in the regulation of energy balance and weight maintenance. It is well described that hypothyroidism is first reflected by weight variation in rodents and humans (Song et al., 2011, doi: 10.1007/s13238-011-1046-x; Mullur et al., 2014, doi: 10.1152/physrev.00030.2013). A large number of studies reported weight loss in mice and rats treated with PTU or MMI starting from 2 weeks (Decherf et al., 2010, doi:10.1073/pnas.0905190107; Chaalal et al., 2014 (reference 22 in the manuscript); Herwig et al., 2014, doi: 10.1089/thy.2014.0169), a well described phenotype that has been associated to hypothyroidism, and what is more, congenital hypothyroidism is characterized by dwarfism in human and mice. However, in our study, as mentioned in material and methods section, euthyroid C57BL/6J and WSB/EiJ mice exhibited an increase of their body weight in a similar way throughout the treatment, reflecting normal growth period and well-being status of young euthyroid adult mice. Also, in a former study, as explained in the manuscript lines 523, we followed weight gain of these two strains starting from the age of 1 month to 4 months of age. We observed that the weight gain curves were similar during this period in both strains under control conditions (Terrien et al., 2019, reference 29 in the manuscript), once again proving that the lower circulating T4 levels of the WSB/EiJ mice (still in the reference range as explained earlier) were not inducing signs of hypothyroidism.
3. Add analysis of serum T3 concentrations.
We agree with the reviewer that T3 serum concentration could have been measured in animals treated with PTU to establish circulating hypothyroidism. Nevertheless, it is well established for years by many studies from our labs and others that PTU treatment reduces both T4 and T3 concomitantly.
Indeed, PTU treatment is a well-established and worldwide-recognized protocol to induce hypothyroidism in rodent, and the dosage of TH levels is only a formality to verify that the treatment has indeed been effective. For example, among many, Cortes and collaborators reported that PTU treatment for 20 days induce a significant decrease level of serum T3 and T4 levels (reference 17 in the manuscript), as also shown by Sener et al., 2006 (reference 38 in the manuscript): in the PTU-treated rats the serum T3 and T4 levels (20.8 ± 0.22 and 0.31 ± 0.01 ng/dl) were found to be significantly lower than those of the control group (55.3 ± 0.63 and 6.810.32 ng/dl, p<0.001). Other authors also attest
circulating hypothyroidism testing only T4, as Artis et al., who reported that serum T4 levels in comparable PTU treated rats were lower than the control values (0.248 ± 0.038 μg/dl vs 1.276 ± 0.578 μg/dl, (reference 21 in the manuscript). Vallortigara et al., in 2008 (reference 39 in the manuscript)
confirmed hypothyroidism by a drastic reduction in T4 levels (by -96%) in PTU/MMI-treated mice compared with controls. A comparable decrease in free-T4 levels in serum was observed in thyroidectomized rats Alzoubi et al., 2009 (reference 23 in the manuscript). Therefore, based on these numerous data, we only assayed T4 to verify that the effect of PTU treatment had indeed induced
circulating hypothyroidism in our treated mice of both strains. Nevertheless, in the present work, we also observed that the weight gain was inhibited by PTU treatment as body weight gain of both strains was decreased throughout the weeks of treatment, as reported by other authors cited above. These data
strengthen the efficient of PTU treatment known to arrest growth of mice as described previously.
To justify the fact that we did not assayed T3 levels, we added a paragraph with the above explanations in the manuscript (lines 339 to 355).
4. L 329-335
“Knowing that T3 is the main transcriptionally active TH, we assessed the the hippocampal T3 availability by measuring main actors in TH signalling as well as the regulation of T3 target genes, as already widely described in the literature [33-35,38]. In the hippocampus, we observed a decrease in T3 target gene expression only in the C57BL/6J mouse strain. The local action of iodothyronine deiodinases contributes to about 50% of T3 produced in postnatal and adult rodents [40].”
Insert the reference 39.
Sorry for the inconvenience. We have now checked the reference list throughout the manuscript and corrected accordingly.
5. L 353-408
This paragraph is too long. Divide it to two paragraphs or more.
The paragraph has been divided in three paragraphs as requested.
6. Section “4.1. Animals, treatment, and sample collections”
How many mice were used in this study? Add the information in the revised manuscript.
A total of max 12 animals/group were used in this study. This was added in this section as requested.
Reviewer 2 Report (New Reviewer)
ijms-1895532
A fine regulation of the hippocampal thyroid signalling protects hypothyroid mice against glial cell activation.
This is well presented and written paper dealing with a very relevant topic and with many implications for neurodegenerative diseases. The applied methodology is appropriate and correctly used. The results are significant providing a series of interesting conclusions, in accordance with the results obtained.
Some minor revisions could be considered:
Line 202: realized/performed.
Lines 335-338: The authors should give a better explanation of the sentence.
-Lines 463-466. The authors should improve the sentence:
may be” , “would have” in the same paragraph?
Perhaps it would be interesting to include a list of abbreviations.
References: please, check “el al.“ in every cite.
Author Response
Some minor revisions could be considered:
Line 202: realized/performed.
This point has been modified in the revised manuscript.
Lines 335-338: The authors should give a better explanation of the sentence.
We apologize for this unclear sentence. It has been rewritten as follow:
“In the hypothyroid C57BL/6J mouse strain, Dio2 mRNA level was unchanged in the hippocampus, and Dio3 mRNA level was reduced. Dio2, as an activating deiodinase, favors the presence of T3 by converting T4 in T3, whereas Dio3 reduces T3 levels, converting T3 in T2. Thus, our results suggest a decrease in T3 local availability in the hippocampus, as Dio2 expression, as not increased, would not allow the compensation of the lower T4 levels entering the cells, due to the circulating hypothyroidism. The decrease in Dio3 expression also confirms that T3 levels are very low in these cells”. (Lines 361-368)
Lines 463-466. The authors should improve the sentence:
may be”, “would have” in the same paragraph?
We acknowledge this fact. We have modified the sentence accordingly:
“Thus, although it is necessary to develop WSB/EiJ strain-specific cognitive assays, all these data allow us to hypothesize that circulating hypothyroidism would have no effect on neuron-glia interaction and thus on hippocampal functions”. (Line 498)
Perhaps it would be interesting to include a list of abbreviations.
According to the IJMS instruction for authors, the abbreviations have to be defined the first timethey are used, but an abbreviation list is not requested.
“Acronyms/Abbreviations/Initialisms should be defined the first time they appear in each of three sections: the abstract; the main text; the first figure or table. When defined for the first time, the acronym/abbreviation/initialism should be added in parentheses after the written-out form”.
References: please, check “el al.“ in every cite.
Sorry for the inconvenience. We have now checked all the references and corrected accordingly.
Round 2
Reviewer 1 Report (Previous Reviewer 1)
Journal: IJMS (ISSN 1422-0067)
Manuscript ID: ijms-1895532-peer-review-v2
Type: Article
Title: A fine regulation of the hippocampal thyroid signalling protects hypothyroid mice against glial cell activation.
Section: Molecular Endocrinology and Metabolism
Special Issue: Molecular Mechanism of Hypothyroidism
This manuscript aims to study relationship between hypothyroidism in mice and hippocampal glial cell activation, neuroinflammation as well as learning and memory deficits. The quality of manuscript has improved during revisions. However, there are still some issues need to be addressed. The authors did not make tracked changes during the revision, thus it is very difficult for the review to check the revised version. For several issues, the authors only responded to the reviewer in the letter, but did not make revisions in the manuscript, so the readers cannot read it. The detailed suggestions and comments are as follows:
1. Introduction
Please add the following sentence.
WSB/EiJ are genetically distinct from other mouse strains, including C57BL/6J, and possess genomic variations (new alleles) that could be the source of the observed phenotypic divergences, as lower body weight and resistance to diet-induced obesity.
2. Lines 79-115
This paragraph is too long. Divide it to two paragraphs or more.
3. Results
Please add some sentences to explain that WSB/EiJ mice are euthyroid and the definition of euthyroid.
4. Lines 274-307
This paragraph is too long. Divide it to two paragraphs or more.
5. Discussion
Please add some text to discuss why you did not analyze serum T3 concentrations.
6. Lines 329-380
This paragraph is too long. Divide it to two paragraphs or more.
7. It is the authors’ responsibility to present their best work to the readers. Please carefully check the entire manuscript, figures, tables and supplementary files.
Author Response
This manuscript aims to study relationship between hypothyroidism in mice and hippocampal glial cell activation, neuroinflammation as well as learning and memory deficits. The quality of manuscript has improved during revisions. However, there are still some issues need to be addressed. The authors did not make tracked changes during the revision, thus it is very difficult for the review to check the revised version. For several issues, the authors only responded to the reviewer in the letter, but did not make revisions in the manuscript, so the readers cannot read it. The detailed suggestions and comments are as follows:
We apologize for the fact that the reviewer did not get the tracked changes nevertheless our revised manuscript was submitted with tracked changes, perhaps they were lost during the processing. To avoid this drawback, we now highlight changes in the text.
Moreover, we do not understand to what refers the sentence “For several issues, the authors only responded to the reviewer in the letter, but did not make revisions in the manuscript,”: in the last answer to reviewers, we always tried to add a sentence in the manuscript in response to the reviewer comments, giving the line where the sentence was added, as reported in the former response to the reviewer. We apologize if this was not clear enough.
- Introduction
Please add the following sentence.
WSB/EiJ are genetically distinct from other mouse strains, including C57BL/6J, and possess genomic variations (new alleles) that could be the source of the observed phenotypic divergences, as lower body weight and resistance to diet-induced obesity.
As requested, this sentence was added lines 87 to 89
- Lines 79-115
This paragraph is too long. Divide it to two paragraphs or more.
As requested, this long paragraph is now divided into 3 paragraphs.
- Results
Please add some sentences to explain that WSB/EiJ mice are euthyroid and the definition of euthyroid.
For the authors, explaining why WSB/EiJ mice are considered euthyroid is part of the discussion, not of the results, this is the reason why this explanation was already present in the discussion section (lines 336 to 347), and this explanation was completed after the first run of revision, as requested by the reviewer:
“Despite the lower T4 circulating levels of WSB/EIJ compared to the C57BL/6J mice, this strain is still considered as euthyroid as its T4 levels is included in the euthyroid reference range for mice [2.6-9 µg/dL] established by the comparison of 28 non-pathological mouse strains (Donahue 1 study https://phenome.jax.org/measures/11531). If we do the analogy with Human, a T4 level is considered normal when it is between 5.0 to 12.0 μg/dL. It is therefore not abnormal to have a variation of two-fold between two different individuals, and that does not mean that the individual with the lower level is hypothyroid, as long as it is inside the normal range. Moreover, as described in our former study, young euthyroid WSB/EiJ mice presented an increase of their body weight similarly to C57BL/6J mice from one month to 4 months of age, reflecting a normal growth period and well-being status of young euthyroid adult mice [29].
To answer the reviewer present request, a reference to these sentences has been added in the results section, line 126.
- Lines 274-307
This paragraph is too long. Divide it to two paragraphs or more.
As requested, this long paragraph is now divided into 3 paragraphs.
- Discussion
Please add some text to discuss why you did not analyze serum T3 concentrations.
As already written in the response to reviewer round 1, we justified the fact that we did not assayed T3 levels, by adding a paragraph with the above explanations in the round 1 revised manuscript (lines 351 to 367 see below):
“It is well established for years by many studies from our labs and others that PTU treatment reduces both T4 and T3 concomitantly. For example, Cortes and collaborators reported that PTU treatment for 20 days induce a significant decrease level of serum T3 and T4 levels [17] as also shown by Sener et al, 2006 [38]: in the PTU-treated rats the serum T3 and T4 levels (20.8 ± 0.22 and 0.31 ± 0.01 ng/dl) were found to be significantly lower than those of the control group (55.3 ± 0.63 and 6.810.32 ng/dl, p<0.001). Other authors also attest circulating hypothyroidism testing only T4, as Artis et al., who reported that serum T4 levels in comparable PTU treated rats were lower than the control values (0.248 ± 0.038 μg/dl vs 1.276 ± 0.578 μg/dl, [21]. Vallortigara et al., in 2008 confirmed hypothyroidism by a drastic reduction in T4 levels (by –96%) in PTU/MMI-treated mice compared with controls [39]. A comparable decrease in free-T4 levels in serum was observed in thyroidectomized rats [23]. In the present study, we confirmed a drastic reduction in T4 levels in both PTU-treated mouse strains as compared to controls, thus validating the circulating hypothyroidism.”
However, to satisfy reviewer requirement, we slightly modified the last sentence, which is now (lines 362 to 365):
“Therefore, based on these numerous data, in the present study we only assayed T4 and not T3 to verify that PTU treatment had induced circulating hypothyroidism in our treated mice of both strains. Indeed, we confirmed a drastic reduction in T4 levels in both PTU-treated mouse strains as compared to controls, thus validating the circulating hypothyroidism.”
- Lines 329-380
This paragraph is too long. Divide it to two paragraphs or more.
As requested, this long paragraph is now divided into 3 paragraphs.
- It is the authors’ responsibility to present their best work to the readers. Please carefully check the entire manuscript, figures, tables and supplementary files.
We carefully checked the whole manuscript and made some corrections when required.
This manuscript is a resubmission of an earlier submission. The following is a list of the peer review reports and author responses from that submission.
Round 1
Reviewer 1 Report
Journal: IJMS (ISSN 1422-0067)
Manuscript ID: ijms-1715512
Type: Article
Title: Hippocampal hypothyroidism in mice promotes glial cell activation and spatial memory deficits
Section: Molecular Endocrinology and Metabolism
Special Issue: Molecular Mechanism of Hypothyroidism
This manuscript aims to study relationship between hippocampal hypothyroidism in mice and glial cell activation, neuroinflammation as well as spatial memory deficits. However, T4 and T3 concentrations in the hippocampus were not measured. Also, relationship of glial cell activation and neuroinflammation as well as spatial memory deficits were not so clear. I have the following comments and suggestions for the authors to improve the quality of the manuscript.
- The title is hippocampal hypothyroidism in mice, however, T4 and T3 concentrations in the hippocampus were not measured.
- Study design
“We induced hypothyroidism in adult C57BL/6J and wild-derived WSB/EiJ male mice by a seven-week propylthiouracil (PTU) treatment. We previously showed that WSB/EiJ mice were resistant to high-fat diet (HFD)-induced obesity, showing no neuroinflammatory response through adaptive abilities, unlike C57BL/6J.”
Why WSB/EiJ mice were resistant to high-fat diet (HFD)-induced obesity, showing no neuroinflammatory response through adaptive abilities? What are the differences between WSB/EiJ mice and C57BL/6J?
- Section “1. Introduction”
Lines 38-43
“Circulating T4 reaches the brain by crossing the blood-brain barrier via specific transporters. Then, T4 enters into the astrocytes via transporters, where it is deiodinated by DIO2 to produce T3 (reviewed in [4]). Subsequently, T3 is released from the astrocytes and capture by neurons via distinct transporters. Neurons are not able to directly generate T3 because they do not express DIO2. Though, they are dependent upon astrocytes for their T3 supply [3,5].”
“Circulating T4 reaches the brain by crossing the blood-brain barrier via specific transporters”, can circulating T3 reaches the brain by crossing the blood-brain barrier via specific transporters?
- Section “4. Materials and methods”
Lines 443-446
“Hypothyroidism was induced in 8-weeks old male mice by given an iodine deficient diet supplemented with 0.15% propylthiouracil (PTU, Envigo Teklan, Madison, WI, USA) for 7 weeks as described in [24].”
“diet supplemented with 0.15% propylthiouracil (PTU”, is the concentration of PTU is 1.5 g PTU/kg diet? Please add the data in the revised manuscript.
- Lines 448-449
“Moreover, PTU also inhibits deiodinase 1, which produces T3 by deiodination of T4 in peripheral tissues, such as blood, liver, or kidney [62].”
Deiodinase 1 produces T3 by deiodination of T4. However, in lines 38-43, the authors wrote that T4 is deiodinated by DIO2 to produce T3. Deiodination of T4 to produce is performed by only DIO1, only DIO2 or both? Does PTU inhibit only DIO1, only DIO2 or both? Please write clearly in the revised manuscript.
- Section “4.2. Circulating thyroid hormone levels”
Lines 454-456
“Serum total T4 concentrations were measured using ELISA kit (Labor Diagnostika Nord (LDN), Nordhorn, Germany) according to the manufacturer’s instruction.”
Please add data of T3.
- Section “2. Results”
Lines 100-101
“Euthyroid WSB/EiJ mice depicted a lower circulating total T4 [3.4 µg/dL] than C57BL/6J mice [6.1 µg/dL] (p= 0.0003).”
WSB/EiJ control mice depicted a lower circulating total T4 [3.4 µg/dL] than C57BL/6J control mice [6.1 µg/dL] (p= 0.0003). I do not think WSB/EiJ control mice are euthyroid.
- Please add experiments on detection of T4 and T3 concentrations in the hippocampus.
- Section “2.2. Consequences of hypothyroidism on hippocampal inflammation”
Lines 136-138
“Regarding astrocytes, we first measured by western-blot the expression of STAT3 and its phosphorylated form (p-STAT3), an inflammatory marker (Figure 3a).”
“Regarding astrocytes”, STAT3 is only expressed in astrocytes?
- Lines 175-176
“We next examined the expression of hippocampal IBA1 and CD11B proteins, as markers of microglia.”
IBA1 and CD11B proteins are markers for inflammation?
- Lines 237-238
“these results reflected an anxiety-like behavior of WSB/EiJ mice that is reduced by PTU treatment.”
These results are so strange. Why?
- Lines 283-285
“Regarding the WSB/EiJ strain, the results we obtained during the acquisition phase of functional assays showed that spatial memory cannot be tested with the Barnes test (Figure A1).”
Where is Figure A1?
- Lines 402-405
“Literature data reported a clear correlation between RC3 level, a brain-specific gene involved in synaptic plasticity and memory via the modulation of Ca2+/calmodulin-dependent signalling [55], and the cognitive function.”
Please add experiments on RC3 and Ca2+/calmodulin-dependent signalling.
- Please draw a figure to show the results and conclusions, the relationship between hippocampal hypothyroidism in mice and glial cell activation, neuroinflammation, as well as spatial memory deficits.
Reviewer 2 Report
The authors report that hippocampal hypothyroidism promotes glial cell activation and spatial memory deficits in mice. To do this, they combine the analysis on C57BL/6J and WSB/EiJ, since they previously demonstrated that WSB/EiJ mice were resistant to high-fat diet-induced obesity, showing no neuro-inflammatory response. The work is interesting and well organized, however major and minor concerns emerge.
Major concerns:
- The weakest point of the work is the failure of the Barnes maze test in WSB/EiJ mice, which in any case should be presented in figure 6. To link hypothyroidism, neuroinflammation, and memory performances the authors should find a memory test useful also for these mice as well. An alternative could be the novel object recognition test, and/or the Morris water maze test. In the absence of this data, all the work done on WSB/EiJ mice is useless. Please, insert the necessary memory test, or remove the data on WSB/EiJ mice, or try to exhaustively explain this aspect.
- To evaluate the anxiety-related behavior (as reported on page 8, line 228) the elevated plus-maze test is necessary. The time spent in the center of the arena during the open field could help the interpretation of the results, but not replace it. Also, why do the authors present the normalized value of Figure 5b? The right data that they should present are the traveled distance (cm) and the time spent in the center (s). If there is a reason to normalize the data they should explain it.
Minor concerns:
- Affiliation number 3 is missing.
- The results are already indicated at the end of the introduction section, why?
- In Figures 3 and 4, the authors should show the representative western blot images of the internal reference sample. Is it actin?
- In the open field test, the total traveled distance and the velocity represent the same result. Why do the authors indicate both?
- The authors indicate that body weight was monitored twice a week during the treatment. Why then don't they show the datum?
- No information on tissue sampling was indicated. How do they take serum and hippocampus? How do they sacrifice the animal? Also, ethical approval (i.e. number, reference) is not reported.
- Why do the authors point to the Barnes maze test statistic (page 15, line 572) outside the Statistical analyses paragraph (page 16, line581)?
- Text formatting problems arise.
Author Response
Reviewer 2
The authors report that hippocampal hypothyroidism promotes glial cell activation and spatial memory
deficits in mice. To do this, they combine the analysis on C57BL/6J and WSB/EiJ, since they previously
demonstrated that WSB/EiJ mice were resistant to high-fat diet-induced obesity, showing no neuroinflammatory response. The work is interesting and well organized, however major and minor concerns
emerge.
Major concerns:
The weakest point of the work is the failure of the Barnes maze test in WSB/EiJ mice, which in any case should be presented in figure 6. To link hypothyroidism, neuroinflammation, and memory performances the authors should find a memory test useful also for these mice as well. An alternative could be the novel object recognition test, and/or the Morris water maze test. In the absence of this data, all the work done on WSB/EiJ mice is useless. Please, insert the necessary memory test, or remove the data on WSB/EiJ mice, or try to exhaustively explain this aspect.
The remark of referee 2 is pertinent and the behavioural tasks mentioned (Morris water maze and object recognition in spatial version) are indeed the most usual tests to analyze spatial memory as previously published in other studies from our lab [23-27]. However, even if the WSB/EiJ mice seem to be a good model to study, homeostatic mechanisms which could explain the resistance to neuroinflammatory processes, they are very difficult to handle for behavioural experiments, in contrast to C57BL/6J mice, even after several sessions of handling. They avoid as much as possible the experimenter, and they always attempt to escape to hide. Thus, we thought that the Barnes maze would be the best choice of test, rather than Morris water maze, as it is known to be more stressful. In particular, we were worried that the WSB/EiJ mice could not handle this stress and even died, as we had already experienced sudden
death in other stressful situations. Moreover, if they would not die, they would probably quickly get tired of swimming, and, given their anxiety-like behaviour, it is possible that they may not have found the platform, swimming mainly in periphery corridor (thigmotaxis). However, the Barnes maze was unfortunately not successful: WSB/EiJ mice were just looking for a hide, without developing any spatial memory strategy.
Regarding the object recognition in spatial version, we had carried out some preliminary experiments in euthyroid WSB/EiJ male mice. Unfortunately, WSB/EiJ mice were too active to really explore objects, and during the retention session, they were not spending more time exploring the moved object.
Thus, we discontinued this test. Finally, as mentioned in the manuscript, in literature we found only one study in which a type of spatial memory has been assessed with these mice (spatial working memory)
[28] (reference 60 in the manuscript). The authors used the Y maze (including specially made covers to minimize escape attempt) and they showed that WSB/EiJ mice have no preference between both arms (familiar and new arms), probably due to their high level of activity. According to the reviewer’s
comment, data obtained in Barnes maze with WSB/EiJ mice have been removed (and noted as data not shown in the manuscript (line 299).
To evaluate the anxiety-related behavior (as reported on page 8, line 228) the elevated plus-maze test is necessary. The time spent in the center of the arena during the open field could help the interpretation of the results, but not replace it. Also, why do the authors present the normalized value of Figure 5b? The right data that they should present are the traveled distance (cm) and the
time spent in the center (s). If there is a reason to normalize the data they should explain it.
We analysed the time in the central, aversive zone of the open-field to detect potential anxiety-like behaviours. However, we agree with the remark from referee 2 that this result could be confirmed by another task. However, the WSB/EiJ mice were not always cooperative, as described above (even after
handling period, 10 min/each day for 7 days) and were very active. In general, they immediately jump out mazes, if they are open. So, in these conditions, the use of elevated plus maze was not possible. To test anxiety, we could also have used the dark/light box but this has already been published [29]. In this
study, the percent of time in light box was 61.13 ± 7.83 % for male C57BL/6J, against 46.89 ± 3.13 % for male WSB/EiJ mice (table 2), suggesting a more anxious behaviour in WSB/EiJ mice. Finally, Kollmus et al 2020 also showed that, even if WSB/EiJ mice are not more active than C57BL/6J in their study, the percent of time spent in open-field center for WSB/EiJ is lower than C57BL/6J mice, in agreement with our results.
Concerning normalizing data, we first expressed the relative time spent or the distance run in the center zone of the open field arena as percentage of total time or distance, to avoid biased conclusions due to the differences in general activity among groups [30]. Then, we normalized each value to the one
obtained for the C57BL/6J CTRL group. This was justified by the variability in general activity among euthyroid groups of both strains, due to the high difference in spontaneous locomotor activity we observed between strains (figure 5a), which would have masked any difference in response to treatment
between strains. Indeed, some authors have described that mouse derived from wild type strains (as are WSB/EiJ) have very different behaviours in the open field test when comparing to laboratory strains as C57BL/6J mice, leading to high variability explained by several factors [31]. To resume, the global
strain effect is so potent that it could mask an eventual difference of response to treatment between strains, thus normalized data can erase the inter-strain variability at basal conditions in order to analyze the difference in response to the treatment between the strains. We include this explanation in the section
of Materials and Methods (4.6.1) (lines 570-573).
Total traveled distance, velocity and distance and time spent in the center zone were recorded.
Represented data were normalized to CTRL C57BL/6J mice in order to avoid biased conclusions induced by the high variability in general activity among strains.
Minor concerns:
Affiliation number 3 is missing.
The third affiliation has been included in the revised version (3 has been changed by §)
The results are already indicated at the end of the introduction section, why?
We followed the instructions for authors concerning the end of the introduction: “Finally, briefly mention the main aim of the work and highlight the main conclusions”
In Figures 3 and 4, the authors should show the representative western blot images of the internal reference sample. Is it actin?
We do not use actin as loading control. The authors apologize for not having been clear enough in the materials and methods section. Actually, for western-blot, we are using Tgx Stain-Free Protein Gels from Biorad, which allow to use stain-free total protein measurement as the loading control, instead of
using actin or other protein as control. In that way, we can ensure that both the target protein and loading control are measured in the linear dynamic range in a typical western blot experiment. Furthermore, stain-free imaging allows for the complete elimination of the inherently problematic use of housekeeping proteins as loading controls on western blots. Stain-free total protein measurement serves as a more reliable loading control than housekeeping proteins, particularly in the loading range commonly used for cell lysates, permitting the user to obtain truly quantitative western blot data by
normalizing bands to total protein in each lane. For further details, see for example [32] (reference 67 in the manuscript). Moreover, regarding the reference to internal control we made in the western-blot section, it refers to the control sample we load in each different experiment to get rid of gel to gel
variation. The reference to the stain free methods is now added in the western-blot section (line 554).
In the open field test, the total traveled distance and the velocity represent the same result. Why do the authors indicate both?
We agree that the total distance and velocity are comparable data. However, we think that velocity is a better indicator that the WSB/EiJ mice are more active than C57BL/6J.
The authors indicate that body weight was monitored twice a week during the treatment. Why then don't they show the datum?
Actually, those measurements were only made as an indicator of the well-being of the animals in an ethical purpose to check for any deleterious effect of the treatment, and were not aimed for analysis. This was now specified in the materials and methods section of the revised manuscript (lines 478-479).
However, if the reviewer estimates that we should delete this sentence, we will do so.
• No information on tissue sampling was indicated. How do they take serum and hippocampus? How do they sacrifice the animal? Also, ethical approval (i.e. number, reference) is not reported.
All these informations are now added to the revised manuscript in the Materials and methods section (4.1) (lines 469-477; 480).
Why do the authors point to the Barnes maze test statistic (page 15, line 572) outside the Statistical analyses paragraph (page 16, line581)?
This was an error, now Barnes maze test statistics have been moved to the Statistical analyses paragraph (lines 619-626).
Text formatting problems arise.
We apologize for those problems, however, it seems that it has been an editing issue as we checked our original file, and the format was coherent all along the manuscript. However, we modified the formatting of the revised manuscript and hope the no new problems would appear.
References
1. A. C. Schroeder and M. L. Privalsky, “Thyroid hormones, T3 and T4, in the brain,” Frontiers in Endocrinology, vol. 5, no. MAR. 2014. doi: 10.3389/fendo.2014.00040.
2. C. P. Marcelino, E. A. McAninch, G. W. Fernandes, B. M. L. C. Bocco, M. O. Ribeiro, and A. C. Bianco, “Temporal pole responds to subtle changes in local thyroid hormone signaling,” J. Endocr. Soc., vol. 4, no. 11, 2020, doi: 10.1210/jendso/bvaa136. (New reference in the manuscript [32])
3. B. Morte et al., “Thyroid hormone-regulated mouse cerebral cortex genes are differentially dependent on the source of the hormone: A study in monocarboxylate transporter-8- and deiodinase-2-deficient mice,”
Endocrinology, vol. 151, no. 5, pp. 2381–2387, 2010, doi: 10.1210/en.2009-0944.
4. V. A. Galton et al., “Thyroid hormone homeostasis and action in the type 2 deiodinase-deficient rodent brain during development,” Endocrinology, vol. 148, no. 7, pp. 3080–3088, 2007, doi: 10.1210/en.2006-
1727.
5. C. K. Glass and J. M. Holloway, “Regulation of gene expression by the thyroid hormone receptor,” BBA - Reviews on Cancer, vol. 1032, no. 2–3. pp. 157–176, 1990. doi: 10.1016/0304-419X(90)90002-I.
6. A. C. Bianco and B. W. Kim, “Deiodinases: Implications of the local control of thyroid hormone action,” J. Clin. Invest., vol. 116, no. 10, pp. 2571–2579, 2006, doi: 10.1172/JCI29812.
7. J. Terrien et al., “Reduced central and peripheral inflammatory responses and increased mitochondrial activity contribute to diet-induced obesity resistance in WSB/EiJ mice,” Sci. Rep., vol. 9, no. 1, pp. 1–19,
2019, doi: 10.1038/s41598-019-56051-4.
8. B. Morte and J. Bernal, “Thyroid hormone action: Astrocyte-neuron communication,” Frontiers in Endocrinology, vol. 5, no. MAY. 2014. doi: 10.3389/fendo.2014.00082.
9. J. C. Davey, M. J. Schneider, K. B. Becker, and V. A. Galton, “Cloning of a 5.8 kb cDNA for a mouse type 2 deiodinase,” Endocrinology, vol. 140, no. 2, pp. 1022–1025, 1999, doi: 10.1210/endo.140.2.6678.
10. A. C. Bianco, D. Salvatore, B. Gereben, M. J. Berry, and P. R. Larsen, “Biochemistry, cellular and molecular biology, and physiological roles of the iodothyronine selenodeiodinases,” Endocrine Reviews,
vol. 23, no. 1. pp. 38–89, 2002. doi: 10.1210/edrv.23.1.0455.
11. M. S. Clerget-Froidevaux, I. Seugnet, and B. A. Demeneix, “Thyroid status co-regulates thyroid hormone receptor and co-modulator genes specifically in the hypothalamus,” FEBS Lett., vol. 569, no. 1–3, pp. 341–345, 2004, doi: 10.1016/j.febslet.2004.05.076.
12. S. Decherf, I. Seugnet, S. Kouidhi, A. Lopez-Juarez, M. S. Clerget-Froidevaux, and B. A. Demeneix, “Thyroid hormone exerts negative feedback on hypothalamic type 4 melanocortin receptor expression,” Proc. Natl. Acad. Sci. U. S. A., vol. 107, no. 9, pp. 4471–4476, 2010, doi: 10.1073/pnas.0905190107.
13. A. Chaalal et al., “PTU-induced hypothyroidism in rats leads to several early neuropathological signs of Alzheimer’s disease in the hippocampus and spatial memory impairments,” Hippocampus, vol. 24, no. 11, pp. 1381–1393, 2014, doi: 10.1002/hipo.22319.
14. G. Şener et al., “Propylthiouracil (PTU)-induced hypothyroidism alleviates burn-induced multiple organ injury,” Burns, vol. 32, no. 6, pp. 728–736, 2006, doi: 10.1016/j.burns.2006.01.002.
15. S. Ashwini, Z. Bobby, and M. Joseph, “Mild hypothyroidism improves glucose tolerance in experimental type 2 diabetes,” Chem. Biol. Interact., vol. 235, pp. 47–55, 2015, doi: 10.1016/j.cbi.2015.04.007.
16. R. Li, K. L. Svenson, L. R. B. Donahue, L. L. Peters, and G. A. Churchill, “Relationships of dietary fat, body composition, and bone mineral density in inbred mouse strain panels,” Physiol. Genomics, vol. 33,
no. 1, pp. 26–32, 2008, doi: 10.1152/physiolgenomics.00174.2007.
17. K. Ceyzériat, L. Abjean, M. A. Carrillo-de Sauvage, L. Ben Haim, and C. Escartin, “The complex STATes of astrocyte reactivity: How are they controlled by the JAK-STAT3 pathway?,” Neuroscience, vol. 330. pp. 205–218, 2016. doi: 10.1016/j.neuroscience.2016.05.043.
18. D. Ito, Y. Imai, K. Ohsawa, K. Nakajima, Y. Fukuuchi, and S. Kohsaka, “Microglia-specific localisation of a novel calcium binding protein, Iba1,” Mol. Brain Res., vol. 57, no. 1, pp. 1–9, 1998, doi: 10.1016/S0169-328X(98)00040-0.
19. A. Karperien, H. Ahammer, and H. F. Jelinek, “Quantitating the subtleties of microglial morphology with fractal analysis,” Frontiers in Cellular Neuroscience, no. JANUARY 2013. pp. 1–34, 2013. doi: 10.3389/fncel.2013.00003.
20. L. H. Duntas and A. Maillis, “Hypothyroidism and depression: Salient aspects of pathogenesis and management,” Minerva Endocrinol., vol. 38, no. 4, pp. 365–377, 2013.
21. J. P. Stohn, M. E. Martinez, and A. Hernandez, “Decreased anxiety- and depression-like behaviors and hyperactivity in a type 3 deiodinase-deficient mouse showing brain thyrotoxicosis and peripheral hypothyroidism,” Psychoneuroendocrinology, vol. 74, pp. 46–56, 2016, doi:10.1016/j.psyneuen.2016.08.021.
22. V. C. Bortolotto et al., “Chrysin reverses the depressive-like behavior induced by hypothyroidism in female mice by regulating hippocampal serotonin and dopamine,” Eur. J. Pharmacol., vol. 822, pp. 78–84, 2018, doi: 10.1016/j.ejphar.2018.01.017.
23. R. Poirier et al., “Deletion of the Coffin-Lowry syndrome gene Rsk2 in mice is associated with impaired spatial learning and reduced control of exploratory behavior,” Behav. Genet., vol. 37, no. 1, pp. 31–50,
2007, doi: 10.1007/s10519-006-9116-1.
24. R. Poirier et al., “Distinct Functions of Egr Gene Family Members in Cognitive Processes,” Frontiers in Neuroscience, vol. 2, no. 1. pp. 47–55, 2008. doi: 10.3389/neuro.01.002.2008.
25. C. Castillon, S. Lunion, N. Desvignes, A. Hanauer, S. Laroche, and R. Poirier, “Selective alteration of adult hippocampal neurogenesis and impaired spatial pattern separation performance in the RSK2-
deficient mouse model of Coffin-Lowry syndrome,” Neurobiol. Dis., vol. 115, pp. 69–81, 2018, doi: 10.1016/j.nbd.2018.04.007.
26. C. Castillon et al., “The intellectual disability PAK3 R67C mutation impacts cognitive functions and adult hippocampal neurogenesis,” Hum. Mol. Genet., vol. 29, no. 12, pp. 1950–1968, 2020, doi:
10.1093/HMG/DDZ296.
27. A. M. Garcia‐Serrano and J. M. N. Duarte, “ Taurine and N ‐acetylcysteine supplementation prevents memory impairment in high‐fat diet‐fed female mice ,” Alzheimer’s Dement., vol. 17, no. S4, 2021, doi: 10.1002/alz.053779.
28. K. D. Onos et al., “Enhancing face validity of mouse models of Alzheimer’s disease with natural genetic variation,” PLoS Genet., vol. 15, no. 5, pp. 1–29, 2019, doi: 10.1371/journal.pgen.1008155.
29. H. Kollmus et al., “A comprehensive and comparative phenotypic analysis of the collaborative founder strains identifies new and known phenotypes,” Mamm. Genome, vol. 31, no. 1–2, pp. 30–48, 2020, doi: 10.1007/s00335-020-09827-3.
30. A. Saoudi, F. Zarrouki, C. Sebrié, C. Izabelle, A. Goyenvalle, and C. Vaillend, “Emotional behavior and brain anatomy of the mdx52 mouse model of Duchenne muscular dystrophy,” Dis. Model. Mech., vol. 14, no. 9, 2021, doi: 10.1242/dmm.049028.
31. A. Takahashi, K. Kato, J. Makino, T. Shiroishi, and T. Koide, “Multivariate analysis of temporal descriptions of open-field behavior in wild-derived mouse strains,” Behav. Genet., vol. 36, no. 5, pp. 763–774, 2006, doi: 10.1007/s10519-005-9038-3.
32. J. E. Gilda and A. V. Gomes, “Western blotting using in-gel protein labeling as a normalization control: Stain-free technology,” Methods Mol. Biol., vol. 1295, pp. 381–391, 2014, doi: 10.1007/978-1-4939-2550-6_27. (New reference in the manuscript [67]).
Round 2
Reviewer 1 Report
Journal: IJMS (ISSN 1422-0067)
Manuscript ID: ijms-1715512-peer-review-v2
Type: Article
Title: Hippocampal hypothyroidism in mice promotes glial cell activation and spatial memory deficits
Section: Molecular Endocrinology and Metabolism
Special Issue: Molecular Mechanism of Hypothyroidism
This manuscript aims to study relationship between hippocampal hypothyroidism in mice and glial cell activation, neuroinflammation as well as spatial memory deficits. The quality of manuscript has improved, but not too much. Authors have only partly replied to the queries raised. For example, T4 and T3 concentrations in the hippocampus were not measured. I have the following comments and suggestions for the authors to improve the quality of the manuscript.
1. The title is hippocampal hypothyroidism in mice, however, T4 and T3 concentrations in the hippocampus were not measured.
I do not agree with the authors that “it is well established and accepted as a standard method in the TH community to estimate TH tissue availability according to T3 transcriptional activity”. The changes of concentrations of THs are not always consistent with changes in transcriptions of TH receptors. I have just read a paper about it. Please read and cite the following paper.
https://doi.org/10.1016/j.scitotenv.2021.145196
If you did not measure T4 and T3 concentrations in the hippocampus, then you cannot use the term “hippocampal hypothyroidism”.
2. Study design
“We induced hypothyroidism in adult C57BL/6J and wild-derived WSB/EiJ male mice by a seven-week propylthiouracil (PTU) treatment. We previously showed that WSB/EiJ mice were resistant to high-fat diet (HFD)-induced obesity, showing no neuroinflammatory response through adaptive abilities, unlike C57BL/6J.”
Why WSB/EiJ mice were resistant to high-fat diet (HFD)-induced obesity, showing no neuroinflammatory response through adaptive abilities? What are the differences between WSB/EiJ mice and C57BL/6J?
I have asked this question last time, but the authors just cited another paper in the response letter, while readers cannot see it. Please make changes in the revised manuscript. Then readers can better understand your paper without need to check other papers.
3. Section “4.2. Circulating thyroid hormone levels”
Lines 483-484
“Serum total T4 concentrations were measured using ELISA kit (Labor Diagnostika Nord (LDN), Nordhorn, Germany) according to the manufacturer’s instruction.”
Please add data of T3. I have made this comment last time, but the authors responded that “in a context of limited quantities of serum, it is more informative to measure T4 than T3 to assess thyroid status”. In fact, T3 is the main form of active thyroid hormone (TH). Also, changes of T4 are not always consistent with changes of T3.
How many mice in a group? To my knowledge, the amount of serum of a mice is enough for detection of concentrations of both T4 and T3 by ELISA kits. If not enough, then the authors did not collect enough blood from the mice. Also, if the authors have difficulties in collection of blood or have used the serum for other analyses, then you should use more mice in a group. There are flaws in the study design.
4. Materials and methods
Section “4.1. Animals treatment, and sample collections”
How many mice in a group? Please insert the information in the revised manuscript.
5. Section “2. Results”
Lines 103-104
“Euthyroid WSB/EiJ mice depicted a lower circulating total T4 [3.4 µg/dL] than C57BL/6J mice [6.1 µg/dL] (p= 0.0003).”
WSB/EiJ control mice depicted a lower circulating total T4 [3.4 µg/dL] than C57BL/6J control mice [6.1 µg/dL] (p= 0.0003). I do not think WSB/EiJ control mice are euthyroid.
The authors responded that “Actually, despite the lower T4 circulating levels of WSB/EIJ compared to the C57BL/6J mice, they are considered as euthyroid and not hypothyroid, as they do not present any signs of hypothyroidism. When compared with 28 other non-pathological mouse strains, as described in [16] (reference 34 in the manuscript), this strain is in the lower of the euthyroid reference range for mice [2.6-9 µg/dL].”
Again, readers cannot see the response letter. Please make changes in the revised manuscript. Please insert the explanations in the revised manuscript.
Also, which signs of hypothyroidism did you observe? Please make changes in the revised manuscript.
6. Lines 297-299
“Regarding the WSB/EiJ strain, the results we obtained during the acquisition phase of functional assays showed that spatial memory cannot be tested with the Barnes test (Data not shown).”
The topic about this manuscript is spatial memory deficits. However, the key results for spatial memory cannot be tested with the Barnes test. This is a large flaw of this manuscript,
7. Figure 5b
Normalized value is shown in Figure 5b. Please present actual detected values.
Reviewer 2 Report
The authors report that hippocampal hypothyroidism promotes glial cell activation and spatial memory deficits in mice. To do this, they combine the analysis on C57BL/6J and WSB/EiJ, since they previously demonstrated that WSB/EiJ mice were resistant to high-fat diet-induced obesity, showing no neuro-inflammatory response. The work is interesting and well organized, however major and minor concerns emerge.
Major concerns:
The weakest point of the work is the failure of the Barnes maze test in WSB/EiJ mice, which in any case should be presented in figure 6. To link hypothyroidism, neuroinflammation, and memory performances the authors should find a memory test useful also for these mice as well. An alternative could be the novel object recognition test, and/or the Morris water maze test. In the absence of this data, all the work done on WSB/EiJ mice is useless. Please, insert the necessary memory test, or remove the data on WSB/EiJ mice, or try to exhaustively explain this aspect.
To evaluate the anxiety-related behavior (as reported on page 8, line 228) the elevated plus-maze test is necessary. The time spent in the center of the arena during the open field could help the interpretation of the results, but not replace it. Also, why do the authors present the normalized value of Figure 5b? The right data that they should present are the traveled distance (cm) and the time spent in the center (s). If there is a reason to normalize the data they should explain it.
Minor concerns:
Affiliation number 3 is missing.
The results are already indicated at the end of the introduction section, why?
In Figures 3 and 4, the authors should show the representative western blot images of the internal reference sample. Is it actin?
In the open field test, the total traveled distance and the velocity represent the same result. Why do the authors indicate both?
The authors indicate that body weight was monitored twice a week during the treatment. Why then don't they show the datum?
No information on tissue sampling was indicated. How do they take serum and hippocampus? How do they sacrifice the animal? Also, ethical approval (i.e. number, reference) is not reported.
Why do the authors point to the Barnes maze test statistic (page 15, line 572) outside the Statistical analyses paragraph (page 16, line581)?
Text formatting problems arise.